# Field testing of a local wind inflow estimator and wake detector

Johannes Schreiber, Carlo L. Bottasso, and Marta Bertelè

Wind Energy Institute, Technische Universität München, 85748 Garching bei München, Germany

**Correspondence:** Carlo L. Bottasso (carlo.bottasso@tum.de)

**Abstract.**

This paper presents the field validation of a method to estimate the local wind speed on different sectors of a turbine rotor disk. Each rotating blade is used as a scanning sensor that, travelling across the rotor disk, samples the inflow. From the local speed estimates, the method can reconstruct the vertical wind shear and detect the presence and location on an impinging wake shed by an upstream wind turbine. Shear and wake awareness have multiple uses, from turbine and farm control to monitoring and forecasting.

This validation study is conducted with an experimental data set obtained with two multi-MW wind turbines and a hub-tall met-mast. Practical and simple procedures are presented and demonstrated to correct for the possible miscalibration of sensors.

Results indicate a very good correlation between the estimated vertical shear and the one measured by the met-mast. Additionally, the proposed method exhibits a remarkable ability to locate and track the motion of an impinging wake on an affected rotor.

## 1 Introduction

Knowledge of the wind turbine inflow can enable several applications. For example, a turbine controller can be improved when scheduled as a function of wind speed (Østergaard et al., 2007). Similarly, a farm controller benefits from knowledge of the atmospheric stability because of its strong effect on wake recovery, and from an improved understanding of wake position (Vollmer et al., 2017) because of its crucial implications on power output and loading. Apart from control applications, other usage scenarios include lifetime assessment and fatigue consumption estimation, which are clearly dictated by the inflow conditions experienced by each turbine (Ziegler and Muskulus, 2016). Moreover, wind farm power and wind forecasting, post-construction site assessment, sector management triggered by wake detection for closely spaced turbines, and estimation of available wind farm power are all additional applications that can profit from knowledge of the inflow affecting each single turbine. Unfortunately, this information is not available on today's wind turbines that, as a consequence, operate "in the dark" based only on a limited awareness of the environment in which they are immersed.

Indeed, turbines are equipped with wind sensors, typically located on the nacelle or the spinner, which are used for aligning the rotor axis into the wind and for identifying whether the cut-in or cut-out wind speeds have been reached. Even though these measurements might be accurate enough for these simple tasks, the actual complexity of the turbine inflow remains completely beyond the reach of such sensors. In addition, wind vanes and anemometers provide pointwise information, while

wind conditions exhibit significant spatial variability not only at the large scale of the farm, as in off-shore plants (Peña et al., 2018) and at complex terrain sites (Lange et al., 2017; Schreiber et al., 2019), but also at the smaller scale of the individual turbine rotor disk (Murphy et al., 2019). More sophisticated measurements can be provided by lidars (Held and Mann, 2019) and other remote sensing technologies, which are however still costly and —being mostly used for assessment, validation and research— are not yet commonly used for production installations.

The concept of using the wind turbine rotor as a wind sensor has been proposed to improve wind condition awareness (Bottasso et al., 2010; Simley and Pao, 2016; Bertelè et al., 2017). In a nutshell, wind sensing uses the response of the rotor —in the form of loads, accelerations and other operational data— to infer the characteristics of the wind blowing on the turbine. Therefore, wind sensing is a sort of model inversion, where the response of the system is used to estimate the disturbance (in this case, the wind). The simplest and probably most widely used wind sensing technique is torque-balance estimation (Ma et al., 1995; Soltani et al., 2013). Thereby, turbine power or torque are used to estimate the rotor-effective wind speed by the power curve or power coefficient. The concept has been more recently extended to estimate other characteristics of the inflow, notably the wind directions and shears, as reviewed in Bertelè et al. (2017).

This paper considers the approach first formulated by Bottasso et al. (2018). Through an aerodynamic "cone" coefficient, this method uses the blade out-of-plane bending moment to estimate the local wind speed at the position occupied by a blade. The method is very similar to the torque-balance estimation of the wind speed, with the important difference that it produces a localized speed estimate instead of a rotor-effective one. The rotating blades therefore operate as scanning sensors that, travelling across the rotor disk, sample the local variability of the inflow. In turn, the local wind speed estimates are used for obtaining two key pieces of information on the inflow: the vertical shear, which is an important load-driver and an indicator of atmospheric stability, and the horizontal shear, which can be used to detect the presence and location of an impinging wake. Today, only a scanning lidar would be able to provide similar information on the inflow, albeit not exactly at the rotor disk —as done here, as the rotor itself is the sensor in this case— and with a very different level of complexity and cost.

The present method has some very interesting features. First, it is model-based, and therefore it does not necessitate of extensive data sets for its training. Second, it is based on an extremely simple model of the rotor (expressed through the cone coefficient), which can be readily computed from a standard aeroelastic model of a wind turbine. Third, the resulting estimator is in the form of a simple look-up-table that is computed off-line, resulting in an on-line on-board implementation of negligible computational cost. Fourth, when load sensors are already installed on the turbine for load-alleviating control or monitoring, this wind sensing technique requires no additional hardware, and therefore its implementation simply amounts to a software upgrade. The wind sensing method considered here has already been tested with blade element momentum (BEM) aeroelastic simulations (Bottasso et al., 2018), large eddy simulations (Schreiber and Wang, 2018), and scaled wind tunnel tests (Campagnolo et al., 2017). Applications related to wake position tracking within a wind farm have been presented in Schreiber et al. (2016) and Bottasso and Schreiber (2018).

Goal of the present paper is to validate the wind sensing approach of Bottasso et al. (2018) in the field. To this end, the method is exercised on a data set obtained with two 3.5 MW turbines, one of which has two blades equipped with load sensors,

and a meteorological mast (met-mast). Since a perfect calibration of the sensors cannot always be guaranteed, another goal of the paper is to present and demonstrate simple and effective methods to correct the measurements and improve accuracy.

The paper is organized as follows. First, the formulation of the wind sensing method is reviewed, including the estimation of rotor-effective and sector-effective wind speeds, as well as of horizontal and vertical shears. Next, the experimental setup is described, including the site layout and the available measurements. The result section represents the core of the paper, and illustrates in detail the performance of the wind sensing technique. A first part of the analysis is concerned with the validation of the vertical shear estimates. Then, the attention is turned to the detection of wake impingement, which is studied by exploiting the waking induced at the site for some wind directions by a neighboring turbine. Finally, the effects of cross-flow are considered, demonstrating that the typical inevitable misalignments between turbine and wind vector do not pollute the estimates. Conclusions and an outlook on future work are given in the last section.

## 2 Methods

### 2.1 Rotor and blade-effective wind speed estimation

Considering a steady and uniform wind speed $V$, the power coefficient $C_\mathrm{p}$ and cone coefficient $C_\mathrm{m}$ (as introduced in Bottasso et al. (2018)) are defined as

$$C_\mathrm{p}(\beta,\lambda,q) = \frac{T_\mathrm{aero}\Omega}{0.5\rho AV^3}, \tag{1a}$$

$$C_\mathrm{m}(\beta,\lambda,q,\psi_i) = \frac{m_i}{0.5\rho ARV^2}, \tag{1b}$$

where $\beta$ is the blade pitch angle, $\lambda = \Omega R/V$ the tip speed ratio, $\Omega$ the rotor speed, $R$ is the rotor radius and $A = \pi R^2$ the swept disk area, $\rho$ is the air density and $q = 1/2\rho V^2$ the dynamic pressure, while $T_\mathrm{aero}$ is the aerodynamic torque. The azimuthal position of the $i$th blade is given by $\psi_i$, while $m_i$ is its out-of-plane root bending moment. Coefficients $C_\mathrm{p}$ and $C_\mathrm{m}$ are readily computed using an aeroelastic model of the turbine, today customarily based on a BEM method that, in the present work, is the one implemented in the FAST code (Jonkman and Jonkman, 2018).

Different approaches to estimate wind speed from the power coefficient are reviewed in detail by Soltani et al. (2013). However, following Bottasso et al. (2018), here we use both the power and the cone coefficients: while the former yields a rotor-effective wind speed (i.e., an average quantity over the entire rotor disk), the latter is used to sample the *local* wind speed at the azimuthal position occupied by a blade. A local radial sampling would require a more sophisticated approach and additional sensors along the blade span, with increased complexity and cost. Given coefficients $C_\mathrm{p}$ and $C_\mathrm{m}$ computed for a reference air density $\rho_\mathrm{ref}$, look-up-tables (LUTs) are generated that return wind speeds given measured loads $T_\mathrm{aero}$ and $m_i$, blade pitch $\beta$, rotor speed $\Omega$ and air density $\rho$. Noting the rotor-effective wind speed estimated from the torque balance

equilibrium as $V_{\mathrm{TB}}$ and the one from blade loads as $V_i$, the inversion of Eqs. (1) yields

$$V_{\mathrm{TB}} = \mathrm{LUT}_{C_{\mathrm{p}}}(\beta, \Omega, T_{\mathrm{aero}}, \frac{\rho}{\rho_{\mathrm{ref}}}), \tag{2a}$$

$$V_i = \mathrm{LUT}_{C_{\mathrm{m}}}(\beta, \Omega, \psi, m_i, \frac{\rho}{\rho_{\mathrm{ref}}}). \tag{2b}$$

Instead of the simple non-linear model inversion adopted here for simplicity, more sophisticated methods can be used, for

example based on Kalman filters or input observers (Soltani et al., 2013), which may slightly improve the results at the cost on an increased complexity. A rotor-effective wind speed can also be obtained from the blade-effective ones by simple averaging over all (three) blades:

$$V_{\mathrm{B}} = 1/3 \sum_{i=1}^{3} V_i. \tag{3}$$

Although in a non-uniform inflow the two rotor-effective speeds $V_{\mathrm{TB}}$ and $V_{\mathrm{B}}$ are not necessarily identical, they are in practice

very similar, as shown later on in the results section. The redundancy offered by $V_{\mathrm{TB}}$ and $V_{\mathrm{B}}$ offers opportunities for sensor calibration, as also described later on.

In Eq. (2a), $T_{\mathrm{aero}}$ is computed from the dynamic torque balance equilibrium $J\dot{\Omega} = T_{\mathrm{aero}} - T_{\mathrm{meas}} - T_{\mathrm{loss}}$, where $J$ is the total rotor, drive-train and generator rotational inertia, while $\dot{\Omega}$ is the rotor acceleration and $T_{\mathrm{meas}}$ is the measured torque at the generator. Mechanical losses in the drive-train are taken into account by the term $T_{\mathrm{loss}}$ (Soltani et al., 2013). Here, for the

accuracy of the wind speed estimate, a dynamic model is used to compute the aerodynamic torque. In fact, the energy converted into rotor acceleration or deceleration is typically large, given the large rotational inertia of the system.

A simpler approach is used for Eq. (2b), where the blade dynamic equilibrium is neglected. This way, the out-of-plane bending moment is directly set to the corresponding measured load, i.e. $m_i = m_{i,\mathrm{meas}}$, where $m_{i,\mathrm{meas}}$ is provided by blade-mounted strain gages, optical sensors or similar devices. The introduction of a flapwise dynamic equilibrium equation, although

certainly possible, would not be straightforward because of the coupling with the tower fore-aft motion and the need to estimate additional relevant modeling parameters. Therefore, in the interest of simplicity and practically applicability, the phase delay caused by the dynamic response of the blade was taken into account by estimating an azimuth bias in the response, as described in §3.7. Due to the high damping of the flap degree of freedom, even the present simplified method seems able to provide accurate results, as also shown in previous simulation studies (Bottasso et al., 2018).

The power and cone coefficients of Eqs. (1) are computed when the rotor axis is aligned with the ambient wind direction. Hence, strictly speaking, Eqs. (2) can be used to estimate wind speeds only in the same aligned conditions. However, this is typically not the case in practice, as turbines are often misaligned with respect to the wind by several degrees. It will be shown later on that moderate misalignments do not significantly affect the estimation of wind speeds, and that the effects of larger misalignments can be corrected for.

## 2.2 Sector-effective wind speed estimation

An average wind speed over a rotor sector can be readily computed by averaging the blade-effective estimate $V_i$ between two azimuthal angles $\psi_a$ and $\psi_b$:

$$V_S = \int_{A_S} V_i(\psi)\,\mathrm{d}A_S, \tag{4}$$

5 where $A_S = (\psi_b - \psi_a)R^2/2$ is the area of the sector. A new sector-effective speed estimate is generated as soon as a blade leaves the sector.

The sector width can be arbitrarily defined. Figure 1 shows the case of the four equally-sized 90-degree-wide sectors used in this work, yielding the four sector-effective wind speed estimates $V_{S,left}$, $V_{S,right}$, $V_{S,up}$, and $V_{S,down}$. Clearly, a finer sampling of the inflow over the rotor disk can be achieved by using smaller sectors. With three blades, each of the sectors is updated 10 three times per rotor revolution. With one single instrumented blade, the update frequency reduces to once per revolution. The effects of sampling frequency on the local wind speed estimates is analyzed in §3.3.

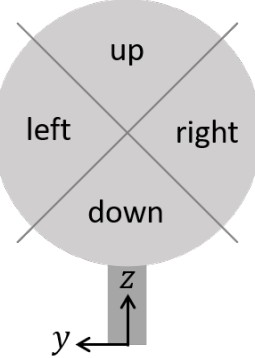

**Figure 1.** Wind turbine rotor disk with sectors and inflow coordinate system. This naming convention is in the downstream viewing direction.

It was shown in Bottasso et al. (2018) that, for a linear inflow shear and a 90-degree-wide sector, the sector-effective wind speed corresponds to the inflow speed at a distance of approximately $2/3R$ from the hub center.

## 2.3 Shear estimation

15 The vertical wind shear is modeled as a power law profile with exponent $\alpha$, while the horizontal shear is assumed to be linear with coefficient $\kappa$. The inflow wind speed $V$ can therefore be written as

$$V(z,y) = V_H \left( \left( \frac{z}{z_H} \right)^\alpha + \kappa \frac{y}{R} \right), \tag{5}$$

where $z$ and $y$ are the vertical and lateral coordinates, respectively, with origin at the turbine foundation, as shown in Fig. 1. Furthermore, $V_H$ is the speed at the hub center, which is located at $z = z_H$ and $y = 0$.

Assuming that the sector-effective speed samples the inflow profile at $\pm 2/3R$ along the $z$ and $y$ axes, according to Bottasso et al. (2018), the shear coefficients can be estimated from the sector-effective wind speeds by using Eq. (5), which yields

$$\alpha_{\mathrm{B}} = \ln\left(\frac{V_{\mathrm{S,up}}}{V_{\mathrm{S,down}}}\right)\left(\ln\left(\frac{z_{\mathrm{H}}+2/3R}{z_{\mathrm{H}}-2/3R}\right)\right)^{-1}, \tag{6a}$$

$$\kappa_{\mathrm{B}} = \frac{3}{2}\frac{V_{\mathrm{S,left}}-V_{\mathrm{S,right}}}{V_{\mathrm{S,left}}+V_{\mathrm{S,right}}}. \tag{6b}$$

This way, the vertical shear is estimated by using the top and bottom sectors, while the horizontal shear by using the two lateral sectors. One could also use all four sectors together, and solve Eq. (5) simultaneously in a least squares sense for both $\alpha_{\mathrm{B}}$ and $\kappa_{\mathrm{B}}$. However, this does not lead to appreciable differences in the results of this paper.

The vertical shear estimate is validated in this work by comparison with an IEC-compliant met-mast, reaching up to hub height. However, shears computed over the whole rotor or over only its lower half can be significantly different; therefore, one

should not compare the full-rotor shear obtained by Eq. (6a) with a lower-half-rotor shear provided by a hub-tall met-mast. To address this issue, a *lower-half-rotor shear estimate* is defined here. This quantity is computed by first averaging the two lateral (left and right) sectors to provide a hub-height speed that, together with the lower sector, is then used to estimate the shear on the sole lower portion of the rotor disk. Using Eq. (5), the lower-half-rotor shear estimate is obtained as

$$\alpha_{\mathrm{lower,B}} = \ln\left(\frac{V_{\mathrm{S,left}}+V_{\mathrm{S,right}}}{2V_{\mathrm{S,down}}}\right)\left(\ln\left(\frac{z_{\mathrm{H}}}{z_{\mathrm{H}}-2/3R}\right)\right)^{-1}. \tag{7}$$

## 3 Results

### 3.1 Experimental setup

This validation study is conducted using an eno114 wind turbine manufactured by eno energy systems GmbH. This turbine, in the following named WT1, has a rated power of 3.5 MW, a rotor diameter $D = 114.9$ m, a hub height $z_{\mathrm{H}} = 92$ m. Two of the blades are equipped with blade load sensors, mounted in close proximity of the root and capable of measuring the two flapwise

and edgewise components.

The site is located approximately 10 km south of the Western Baltic Sea in a slightly hilly terrain without abrupt changes in elevation, approximately 1 km east of the village of Brusow (Germany), as described by Bromm et al. (2018). During the time of the year of the test campaign, the site is characterized by prevailing westerly wind directions, mostly neutral atmospheric stratification, and wind veers between 0 and 10 deg (Bromm et al., 2018).

At the site, a second turbine of the same type, named WT2, and a meteorological mast are also installed. Figure 2 shows a satellite image of the site, including the waking directions and distances among the three installations. WT1 is downstream of the met-mast for a wind direction $\Gamma_{\mathrm{MM->WT1}} = 192.5$ deg, while WT1 is waked by WT2 for $\Gamma_{\mathrm{WT2->WT1}} = 145$ deg. The met-mast is equipped with a wind vane (manufactured by Thies GmbH, catalogue number 4.3150.00.212) installed at $89.4$ m and three cup anemometers (also manufactured by Thies GmbH, catalogue number 4.3351.00.000) at different heights, the

topmost reaching $91.5$ m, which is just half a meter shy of the turbine hub height. The relevant heights of the turbine and met-mast anemometers are shown in Fig. 3.

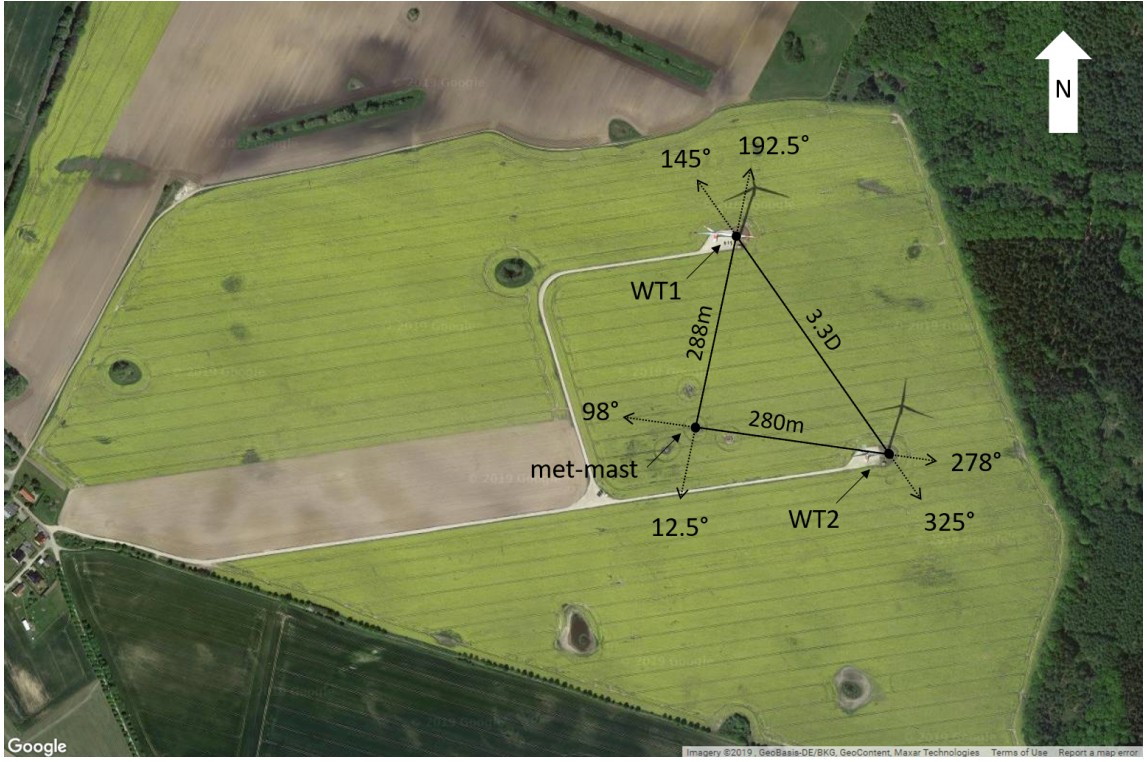

**Figure 2.** Satellite image with WT1, WT2 and met-mast, including waking directions and distances (© Google Maps).

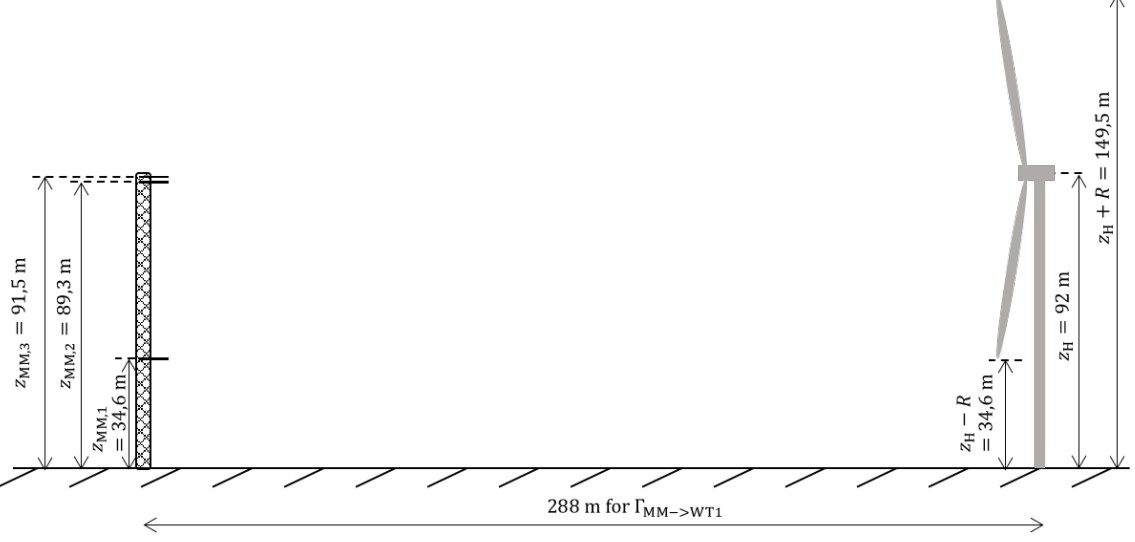

**Figure 3.** Sketch (to scale) of met-mast and WT1 with relevant dimensions.

## 3.2 Measurements

Synchronized measurements of WT1 and the met-mast were made available by the turbine manufacturer and operator for 41 days from October 19 to November 29, 2017. The measurements include main shaft torsion $T_{\mathrm{meas}}$, blade root out-of-plane bending moments for two blades $m_{1,2}$, rotor speed $\Omega$, blade pitch $\beta$, and rotor azimuth position $\psi$. The air density $\rho$ was computed by the ideal gas law using measured air pressure and temperature. Met-mast measurements include wind speed $V_{\mathrm{MM},1-3}$ at the three heights $z_{\mathrm{MM},1-3}$, and wind direction $\Gamma_{\mathrm{MM}}$ at $89.4$ m.

All measurements were sampled at 10 Hz. To eliminate higher frequency turbine dynamics and measurement noise, the rotor speed and torque signals were low pass filtered using a fifth order Butterworth filter with a $-3$ db cut-off frequency of 6 rpm.

The long-term average readings of the two blade load sensors are expected to be equal. However, when comparing the mean sensor values for any of the available days, the relative difference between the two blades was found to be between $4.8$ and $5.8\%$, whereas the absolute differences varied between $-100$ and $-300$ kNm. This mismatch between the two blades suggests a consistent measurement error of one or both sensors. The cause for this error could not be ascertained, but might be due to miscalibration, sensor drift, or pitch misalignment. As an exact determination of the root reason of such inconsistencies is often difficult in a field environment (Bromm et al., 2018), a cause-independent correction method was used here. The first 24 hours of data were used to identify a scaling factor $s = 0.0274$ such that $\overline{m}_1(1 + s) = \overline{m}_2(1 - s)$, where $\overline{(\cdot)}$ indicates a mean value. This scaling factor was then used to correct the sensor readings for the whole data set. For a long term implementation, a similar correction could be applied periodically to compensate time drifts. Notice that this scaling simply ensures consistent measurements between the two sensors, but not their absolute accuracy, which is corrected later in §3.6 by comparison between the rotor-effective wind speeds $V_{\mathrm{TB}}$ and $V_{\mathrm{B}}$. In fact, as these two quantities are based on independent measurements (torque and blade loads), they provide an opportunity to calibrate one or the other sensor.

The data set was filtered, retaining only measurements corresponding to normal turbine operation with pitch and rotor speed within the LUT limits (see §3.5). Measurements taken during yawing manoeuvres were also discarded. In fact, yaw generates additional loads on the blades that would be erroneously interpreted by the observer, resulting in a pollution of the wind estimates. For an observer to accurately estimate wind even during yaw maneuvers, yaw-induced loads could be pre-computed and stored into a look-up table; during operation, one could interpolate within the table in terms of the current yawing rate and possibly wind speed (in case also yaw-induced aerodynamic loads, in addition to the inertial ones, need to be taken into account), and remove the resulting loads from the measured ones. This procedure was however not tested in this work, and therefore yaw maneuvers were eliminated from the data set. After each discarded measurement, an interval of one minute for the estimator re-initialization was accounted for.

The statistical analysis reported below is conducted with 10-minute averages, which are standard in several wind energy applications. However, higher frequency estimates are indeed possible, as shown in §3.7. Of the initial data set, a total of $4,279$ consecutive 10-minute quantities were obtained, representing approximatively 30 days of operation.

### 3.3 Estimator update frequency

The sampling rate of the sector-effective wind estimator varies depending on rotor speed and the number of instrumented blades. For the present case, where only two blades are equipped with load sensors and the rotor speed varies between 5 and 12 rpm, the wind speed estimate update frequency varies approximately between 0.17 and 0.4 Hz. Notice that, since only two out of three blades are instrumented, the update frequency is not constant —even at constant rotor speed.

To quantify the effects of a limited update frequency, Fig. 4 shows the met-mast-measured shear coefficient. The solid black line represents the shear computed based on the signals provided by the cup anemometers at a 10 Hz sampling frequency. The red dashed line reports that same signal downsampled at 0.17 Hz, which is the estimator update frequency for low rotational speeds. A comparison between the two curves shows that even this slowest update frequency is high enough to capture the most energetic fluctuations of the inflow.

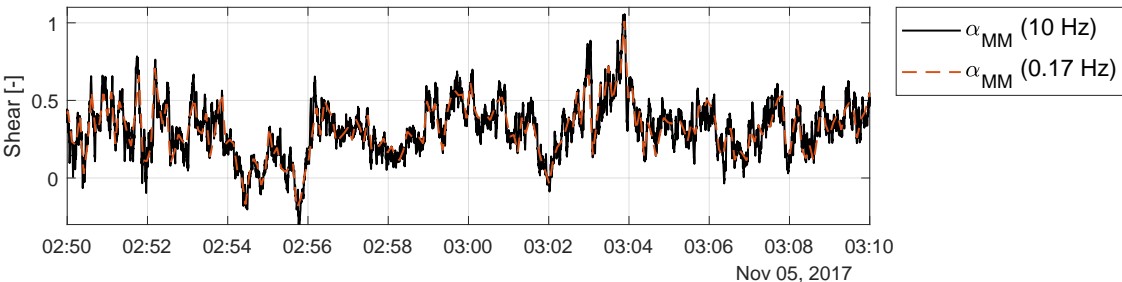

**Figure 4.** Time series of the met-mast-measured shear coefficient, at the original acquisition frequency of the cup-anemometers (10 Hz) and downsampled at 0.17 Hz, which is the sector-effective wind estimation frequency for low rotor speeds.

### 3.4 Reference inflow

The ambient inflow measured by the met-mast is assumed to obey the vertical power law given by Eq. (5). Consequently, the met-mast-measured hub-height reference speed $V_{\mathrm{ref}}$ and power exponent $\alpha_{\mathrm{MM}}$ were computed as best fits of the mast measurements at the three different available heights, i.e.

$$\left(V_{\mathrm{ref}}, \alpha\right) = \arg \min_{V_{\mathrm{ref}}, \alpha} \sum_{i=1}^{3} \left(V_{\mathrm{PL}}(z_{\mathrm{MM},i}, V_{\mathrm{ref}}, \alpha) - V_{\mathrm{MM},i}\right)^2. \tag{8}$$

Only two measurements at two different heights are strictly necessary in order to compute the two parameters of the power law $V_{\mathrm{ref}}$ and $\alpha$. In the present case three measurements are available, although the highest two anemometers, being only about 2 m apart, essentially provide the same information.

Depending on wind direction, the met-mast is located up to 288 meters upstream of WT1, as shown in Fig. 2 for $\Gamma_{\mathrm{MM}->\mathrm{WT1}}$. To synchronize met-mast and turbine measurements, assuming Taylor's frozen turbulence hypothesis, each 10-minute met-mast

measurement was time-shifted by $\Delta t = s_{\mathrm{MM->WT1}}/V_{\mathrm{ref}}$, where $s_{\mathrm{MM->WT1}}$ is the downstream distance from met-mast to WT1.

## 3.5 Look-up-table implementation

An aeroelastic model of the turbine was provided by the turbine manufacturer, implemented in the software FAST (Jonkman and Jonkman, 2018). To compute the power and cone coefficients of Eq. (1), a total of 10,626 dynamic simulations were performed in steady and uniform wind conditions for all combinations of $\beta \in [0:1:20]$ deg, $\Omega \in [3:0.5:14]$ rpm and $V \in [1:1:22]$ m/s, which took just a few hours on a standard desktop PC. Eliminating the tower and drive-train dynamics, a converged periodic response was achieved in three rotor revolutions.

Considering the last revolution, the power coefficient was computed from the mean torque, while the cone coefficient was obtained from the blade root out-of-plane bending moment of one of the blades as a function of $\psi$. The look-up-tables were compiled, for each $\beta$, $\Omega$ and —if applicable— $\psi$, by computing speed as a function of load. If the blade is stalled or partially stalled, the speed-load relationship is non-monotonic. When this happens, the rotor-effective wind speed $V_{\mathrm{TB}}$ of Eq. (2a) can be used to resolve the indeterminacy and identify the correct speed corresponding to the measured load.

## 3.6 Validation of rotor-effective wind speed estimation

First, the rotor-effective speed estimates $V_{\mathrm{TB}}$ (computed through the torque balance equilibrium by Eq. (2a)) and $V_{\mathrm{B}}$ (computed using blade bending moments by Eq. (3)) are compared to each other and to the reference met-mast speed given by Eq. (8). A direct comparison between $V_{\mathrm{TB}}$ and $V_{\mathrm{B}}$ revealed that the latter provides systematically slightly higher wind speeds than the former. This discrepancy may be caused by sensor drift, miscalibration, pitch misalignment and/or deficiencies of the simulation model used to compute the aerodynamic coefficients. Unfortunately, the root causes of the discrepancy could not be determined within the scope of the present work, nor the simulation model could be systematically validated; this is probably the norm rather than the exception also in many practical cases when working in the field. To pragmatically correct these sources of estimation bias, all speed estimates ($V_{\mathrm{B}}$, $V_{\mathrm{S,left}}$, $V_{\mathrm{S,right}}$, $V_{\mathrm{S,up}}$ and $V_{\mathrm{S,down}}$) in the remainder of the paper were scaled by a factor $c = 0.928$. This scaling ensures the best correlation between $V_{\mathrm{B}}$ and $V_{\mathrm{TB}}$, and was identified based on the first seven days of measured data. Note that a direct scaling of the load measurements is also possible and potentially even more accurate.

It is worth pointing out that the redundancy of the two estimates $V_{\mathrm{B}}$ and $V_{\mathrm{TB}}$ offers the opportunity to ensure the consistency between different sets of sensors (the ones measuring blade loads and the ones providing rotor torque). For example, here the torque sensors were properly calibrated, as indicated by the independent measurements of the met-mast, while the blade load sensors were not. Therefore, the redundancy was used to calibrate the load sensors against the torque ones. Similar recalibration procedures might be used also in situations where a met-mast is not available, if one can ensure that at least one set of sensors is properly calibrated.

After correction, a comparison between met-mast reference speed $V_{\mathrm{ref}}$ and torque balance estimates $V_{\mathrm{TB}}$ and $V_{\mathrm{B}}$ is shown in Figs. 5 and 6, respectively. These results include only $3,420$ data points where the met-mast wind direction lies between 180

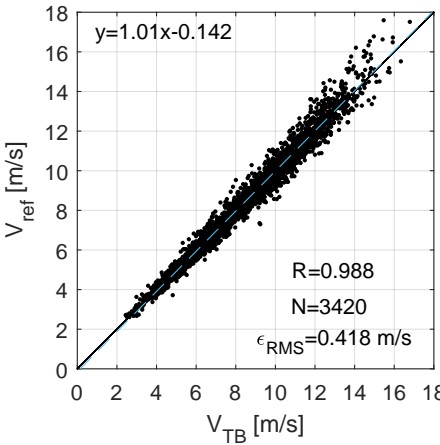

**Figure 5.** Torque-balance-based rotor-effective wind speed $V_{\mathrm{TB}}$ (Eq. (2a)) vs. met-mast reference wind speed $V_{\mathrm{ref}}$ (Eq. (8)).

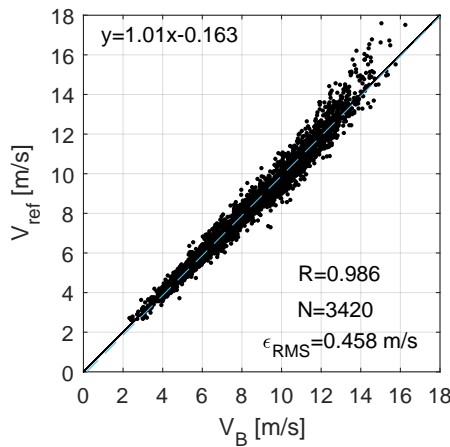

**Figure 6.** Corrected bending-load-based rotor-effective wind speed $V_{\mathrm{B}}$ (Eq. (3)) vs. met-mast reference wind speed $V_{\mathrm{ref}}$ (Eq. (8)).

and 337.5 deg, to avoid conditions where the turbine or the met-mast operate in the wake of either WT1 or WT2 (assuming a $\pm 35$ deg margin). The Pearson correlation coefficient $R$ is approximatively equal to $0.99$, while the root mean squared error is $\epsilon_{\mathrm{RMS}} \approx 0.44$ m/s and the linear best fit ($y = ax + b$) has a slope $a = 1.01$ and an offset $b \approx -0.15$ m/s. These results indicate that, after calibration, the two methods correlate well with the (approximate) ground truth provided by the met-mast, and that

5 both yield very similar estimates.

### 3.7 Validation of vertical shear estimation

After discarding waked conditions from turbine WT2 (with a $\pm 35$ deg margin), an analysis of the long-term mean horizontal shear revealed it to be non-zero. This finding is in contrast with expectations. In fact, while for a narrow wind direction sector some horizontal shear due to local orography or vegetation can be expected, such effects should disappear considering the

10 complete wind rose.

This behavior can be explained by a possible bias in the measurement of the azimuthal position of the rotor, which has the consequence of generating a non-zero horizontal shear and reducing the vertical one. In addition, another effect should be considered: as no blade dynamics were included in the model (see §2.1), the response of the blade is assumed to instantaneously follow a wind speed change. This is in reality not true, and the actual response will have a phase delay, which appears as yet

15 another source of azimuthal bias.

The expected behavior of the horizontal shear can be used for eliminating these effects. In fact, enforcing a null long-term average horizontal shear corrects both for azimuth sensor bias and for having neglected blade dynamics. To this end, the vertical and horizontal shears were rotated by $\psi_{\mathrm{bias}}$, until a null mean horizontal shear was obtained. Accordingly, the mean

vertical shear also reached its maximum. Using again the first seven days of measurements, the azimuth bias was identified as $\psi_{\text{bias}} = 14.8$ deg. In the remainder of this work, the sector-effective wind speeds and the two shears are computed using the corrected azimuth signal $\psi_{\text{corr}} = \psi + \psi_{\text{bias}}$.

The effects of blade dynamics would be more precisely rendered by a rotor-speed-dependent azimuth bias. In fact, by
repeating the shear rotation for binned values of the rotor speed, a clear bias/rotor speed correlation was observed, with bias values in the range between about 12 and 19 deg. In addition, other effects could cause the azimuth bias to drift over time; indeed, a bias of 16.3 deg was found by using the last seven days of data, a slightly different value than the one obtained using the first seven days. However, these slight variations of the bias and its variability with rotor speed have only a very limited effect on the quality of the results. Therefore, for simplicity, it was decided to use the constant average value of 14.8 deg.

As previously discussed, the reference inflow profile measured by the met-mast with Eq. (8) only includes measurements up to hub height. Accordingly, the load-based lower-half-rotor vertical shear $\alpha_{\text{lower,B}}$ (computed by Eq.(7) in terms of the two horizontal and the bottom sectors) is the only shear that can be validated with respect to met-mast measurements.

A 12-hour excerpt from the complete set of results is shown in Fig. 7, where 10-minute means of measurements and estimates are provided as functions of time. Notice that the data points are not equally spaced because of the elimination of yawing
maneuvers and other conditions not accounted for in the LUTs.

The top subplot shows the wind direction $\Gamma_{\text{MM}}$ measured at the met-mast and the turbine yaw orientation $\gamma_{\text{WT1}}$; the direction for which the met-mast is directly upstream of the sensing turbine is $\Gamma_{\text{MM}->\text{WT1}} = 192.5$ deg, and it is shown by a horizontal solid line.

The second subplot from the top shows the reference wind speed $V_{\text{ref}}$ measured at the met-mast, together with the torque-
balance $V_{\text{TB}}$ and blade-load-based $V_{\text{B}}$ rotor-effective speeds. As already noticed, both methods provide very similar results; in addition, especially for wind directions where mast and turbine are nearly aligned, both follow the reference very closely.

The third subplot from the top shows again the met-mast reference wind speed at hub height (solid line) and the one at $z_{\text{H}} - 2/3R$ (dashed line). The respective load-based estimates are indicated with a blue solid line and ● symbols for the hub-height speed, and with a red solid line and × symbols for the lower-height speed. Both estimates correlate well with their
respective references, especially when mast and turbine are aligned. The small rotor icon shows, using the color code of the subplot, the two horizontal sectors (used to estimate the hub height wind speed $1/2(V_{\text{S,left}} + V_{\text{S,right}})$) and the lower sector.

The last subplot finally shows the mast vertical shear $\alpha_{\text{MM}}$ and the load-based estimate $\alpha_{\text{lower,B}}$, computed based on the data shown in the third subplot using Eq. (7). Except for some small underestimation and noise, the load-based shear follows the reference quite accurately. The load-based horizontal shear $\kappa_{\text{B}}$ is also reported in the same figure. Although no met-mast
reference is available in this case, as expected the horizontal shear is always essentially null.

Figure 8 shows the correlation between the lower-half-rotor shears $\alpha_{\text{lower,B}}$ and $\alpha_{\text{MM}}$. Only wind directions from 190 up to 200 deg are included in the figure, resulting in $N = 155$ 10-minute data points. These conditions contain the direction where the met-mast is directly upstream of WT1. The Pearson correlation coefficient is $R = 0.92$. The shear is underestimated with respect to the met-mast reference by a factor $1/a = 0.88$, obtained by the linear best fit ($y = ax + b$) shown in the figure with
a blue dashed line. By looking at the third plot from the top in Fig. 7, a comparison of the wind speed at hub height and at

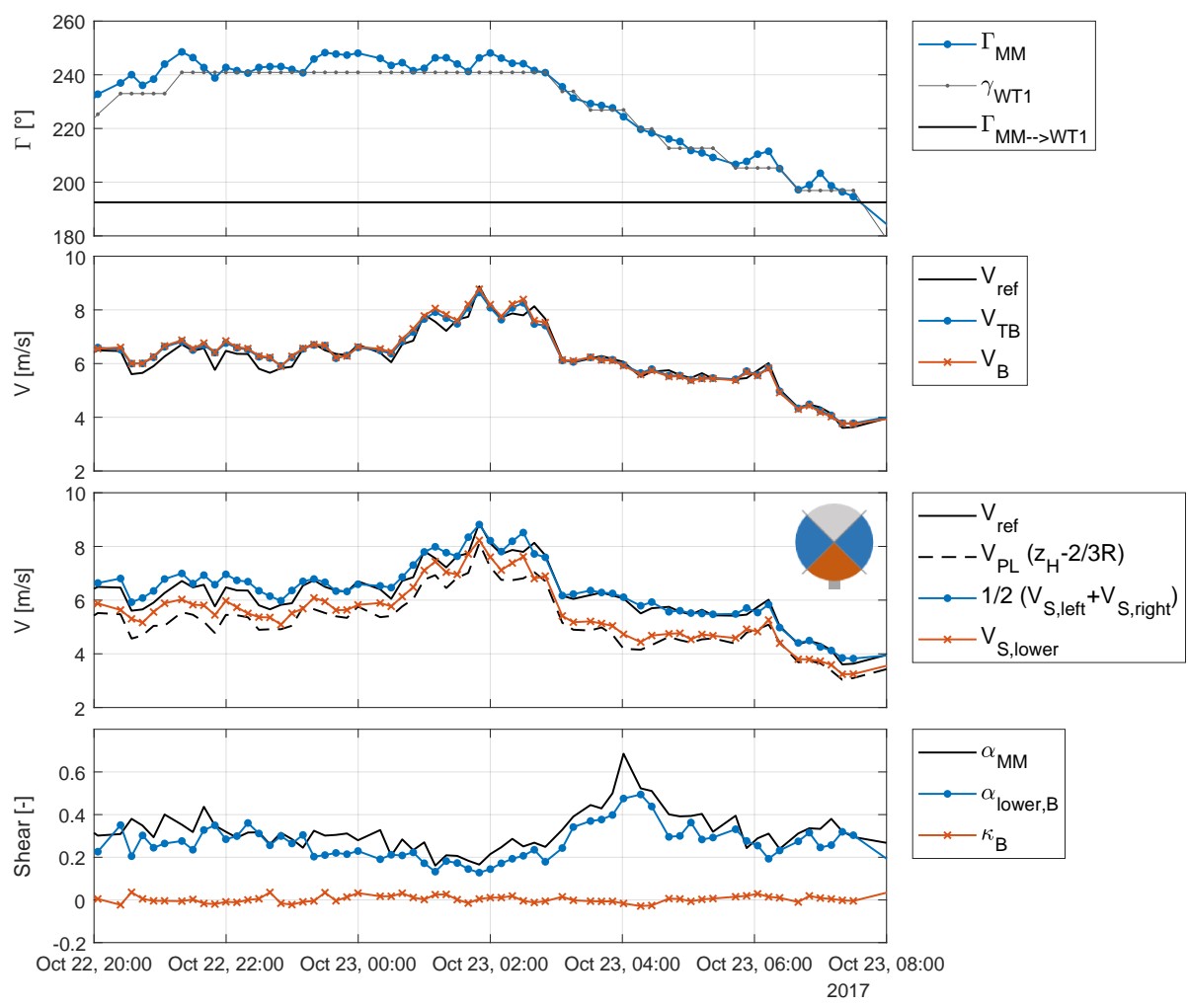

**Figure 7.** Time series reporting met-mast wind direction and turbine yaw orientation (top subplot), met-mast and estimated rotor-effective wind speeds (second subplot), speeds at different heights (third subplot), and met-mast and estimated vertical and horizontal shears (bottom subplot).

$z_{\mathrm{H}} - 2/3R$ with their respective met-mast references indicates that the former is quite accurate, while the latter has a small positive bias. This difference could possibly be caused by a non-ideal power law inflow profile (Møller et al., 2020), leading to a biased met-mast reference shear, although a definitive explanation of this mismatch could not be reached with the present data set. Figure 9 shows the correlation between the full-rotor shear $\alpha_{\mathrm{lower},B}$ and the lower-half-rotor shear $\alpha_{\mathrm{MM}}$. As the two

shears are computed over two different vertical distances, their correlation is lower than in the case of Fig. 8, as expected.

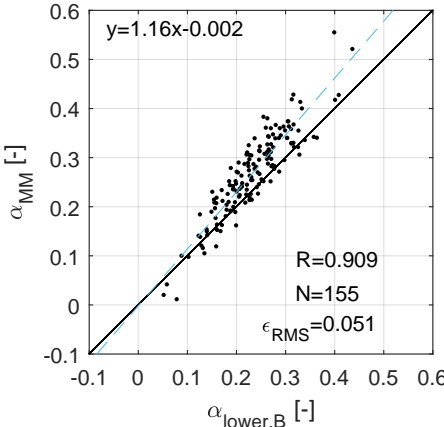

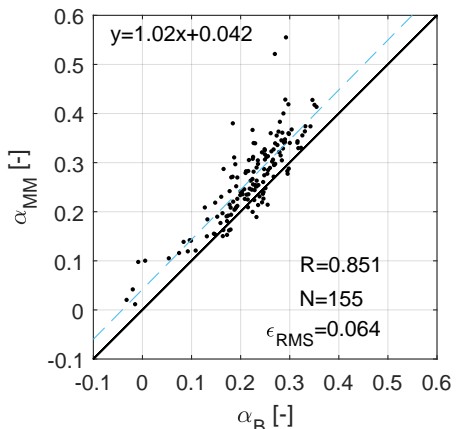

**Figure 8.** Correlation between the lower-half-rotor vertical shear $\alpha_{\mathrm{lower},B}$ and the met-mast shear (up to hub height) $\alpha_{\mathrm{MM}}$, for wind directions from 190 to 200 deg.

**Figure 9.** Correlation between the rotor-equivalent (full rotor) vertical shear $\alpha_B$ and the met-mast shear (up to hub height) $\alpha_{\mathrm{MM}}$, for wind directions from 190 to 200 deg.

A more complete overview of the results, including a broader range of wind directions, is shown in Fig. 10. The $x$-axis reports wind directions from 180 to 340 deg, in 10 deg-wide bins. All results of Fig. 8 fall in the second bin from the left. The number of available measurements $N$ within each bin is shown in the first subplot from the top. The second subplot shows the Pearson correlation coefficient $R$, between the met-mast reference $\alpha_{\mathrm{MM}}$ and the load-based shear estimate $\alpha_{\mathrm{lower},B}$.

Here and in the other plots, a blue solid line indicates results for the lower-half-rotor shear, while a red dashed line is used for the full-rotor shear. The best correlation is achieved for the wind direction where the met-mast is directly upstream of the turbine ($\Gamma_{\mathrm{MM->WT1}} = 192.5$ deg). For the same wind direction bin, also the minimum root mean squared error is achieved, as shown in the third subplot from the top. Considering that all wind directions are for unwaked met-mast and turbine, these results suggest the presence of a spatial shear variation, probably caused by the local vegetation and/or the village in the west

that is partially visible in Fig. 2. This interpretation is also confirmed by the last two subplots, which show the linear best fit coefficients $a$ and $b$. For wind directions up to 235 deg, the slope coefficient $a$ achieves values between 1.02 and 1.18, increasing up to 1.67 in the remaining wind directions. The constant $b$ is nearly zero for all wind direction values.

Looking at the plots, it appears that the full-rotor shear differs from the lower-half-rotor shear, as already reported by Murphy et al. (2019) and as also observed earlier here in Fig. 8. The validation of the full-rotor shear estimated by the proposed method

would necessitate a met-mast reaching the rotor top height or a velocity-azimuth display (VAD) lidar, which however were not

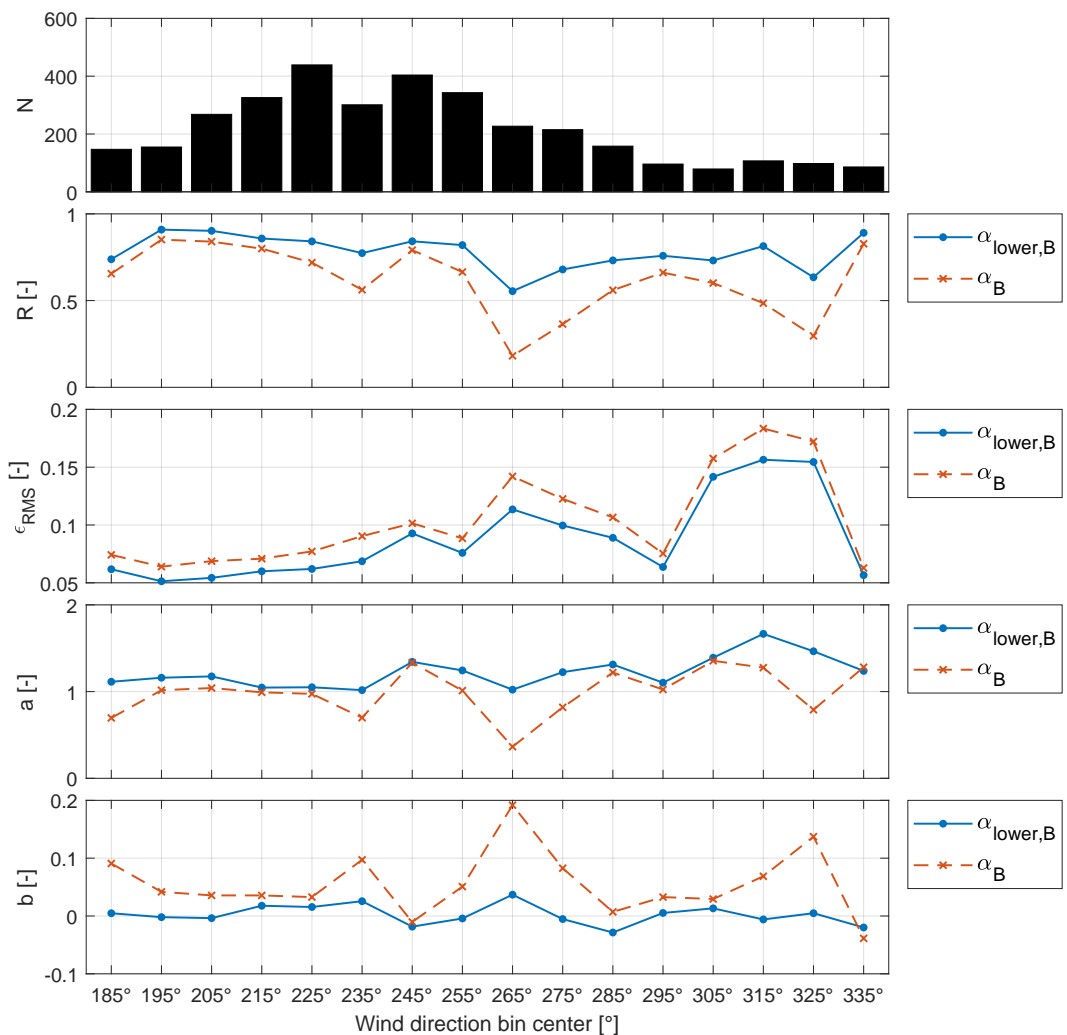

**Figure 10.** Statistics of the shear estimates as functions of wind direction. Blue solid line: lower-half-rotor shear; red dashed line: full-rotor shear. Top subplot: number of 10-minute data points; second subplot: Pearson correlation coefficient; third subplot: root mean squared errors; fourth and fifth subplots: linear best fit coefficients ($y = ax + b$).

available for the present research. Nonetheless, the results obtained for the lower-half-rotor shear appear to be very encouraging, and there is no technical reason why similar results should not be achievable for shear estimates over the entire rotor disk.

Finally, the effects of a higher temporal resolution are considered. Figure 11 compares the 10 Hz lower-half-rotor vertical shear to the met-mast reference; this figure is therefore similar to the last subplot in Fig. 7, which was however obtained with
10-minute averages. Within the 20 minutes considered in the figure, the wind direction was approximately constant and equal to 190 deg, resulting in the met-mast being 2D directly upstream of the turbine, while the wind speed was approximately equal to 7 m/s. Based on the wind speed, the met-mast signal was time-shifted assuming Taylor's frozen turbulence hypothesis. The plot shows that the load-based estimate $\alpha_{\mathrm{lower,B}}$ follows the main trends of the met-mast reference $\alpha_{\mathrm{MM}}$. There are however discrepancies at the higher frequencies. It is not possible to conclusively determine the causes of these differences based
exclusively on the available data. However, the non-colocation of the measurements might clearly be among the reasons. For example, the spike of the met-mast shear at 03:04 is not visible in the load signals, which might indicate a local turbulent fluctuation at one of the met-mast anemometers not rigidly convecting downstream to the turbine rotor.

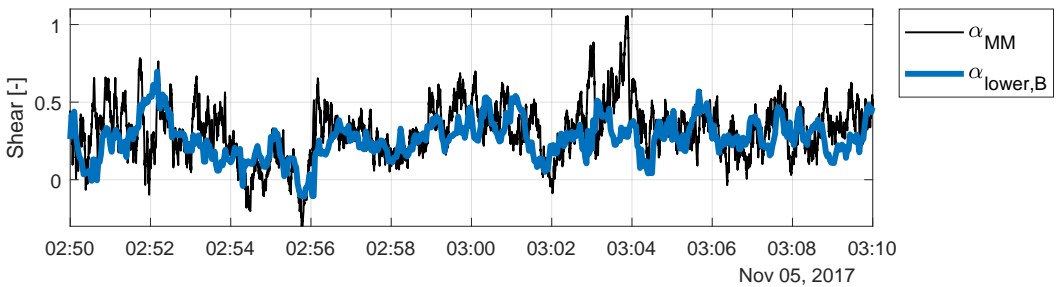

**Figure 11.** Comparison of 10 Hz met-mast vertical shear $\alpha_{\mathrm{MM}}$ with lower-half-rotor shear $\alpha_{\mathrm{lower,B}}$ during a period of 20 minutes.

## 3.8   Validation of wake detection

As no measured reference for the horizontal shear was available for this study, the wake of the second turbine was used for
a qualitative validation. This wake interference study nicely illustrates the very interesting wake detection capabilities of the proposed method.

Figure 12 reports a time series corresponding to 12 hours of operation, which experienced wind direction changes from approximatively 100 to 180 deg. This data subset includes a significant duration where WT1 is waked by WT2. The first subplot from the top shows the met-mast wind direction $\Gamma_{\mathrm{MM}}$ and turbine yaw orientation $\gamma_{\mathrm{WT1}}$, where the waking direction
$\Gamma_{\mathrm{WT2->WT1}}$ is reported as an horizontal solid line (cf. also Fig. 2). The second subplot shows the reference met-mast wind speed $V_{\mathrm{ref}}$, as well as the load-based rotor-equivalent estimates $V_{\mathrm{TB}}$ and $V_{\mathrm{B}}$. The reference 10-minute turbulence intensity $\mathrm{TI}_{\mathrm{ref}}$ computed from $V_{\mathrm{ref}}$ is shown on the right $y$-axis. The third subplot shows the sector-effective wind speeds $V_{\mathrm{S,right}}/V_{\mathrm{ref}}$ and $V_{\mathrm{S,left}}/V_{\mathrm{ref}}$ for the two horizontal sectors, non-dimensionalized by the met-mast reference wind speed. The small rotor

icon shows, using the color code of the subplot, the left (red) and right (blue) sectors. The last subplot reports the horizontal shear estimate $\kappa_{\mathrm{B}}$ computed according to Eq. (6b).

Vertical dashed lines are used to highlight four time instants, labelled with the letters from A to D. For each of these time instants, the position of the wakes of the two turbines is visualized in the lower part of Fig. 12 using the FLORIS wake model (Doekemeijer and Storm, 2019). The yellow color indicates the ambient wind speed, while the blue color is used for the lower speed in the wakes. The rotor disk of WT2 is shown with a solid black line, while a red line is used for the left sector of WT1 and a blue line for the right one. Finally, the small cross symbol indicates the met-mast (MM) position.

At instant A (time equal to 02:05), Fig. 12 shows that the wind direction reaches 130 deg and the left sector of WT1 gets waked by WT2, as clearly illustrated by a reduced speed in the left sector and a negative horizontal shear. At time instant B (02:35), the wind direction has turned back to 122 deg: as the turbine is not waked anymore, the estimated shear is null and an equal wind speed is estimated on both the left and right sectors. The rotor-effective wind speed is slightly smaller than the met-mast reference value; however, for this wind direction, the met-mast is not aligned with the turbine, which might explain this small discrepancy. At time instant C (03:45), the wind direction has increased and WT1 is waked again ($\Gamma_{\mathrm{WT2->WT1}} = 145$ deg): after an initial reduction in the left sector speed, also the right sector is affected (dropping below 0.7), indicating a full waked condition. This is further confirmed by the reduction in the rotor-effective wind speeds with respect to the one measured by the met-mast. Later, a wake impingement on the right sector is observed at time D (05:00), followed by a second full waking at time 05:30. At 06:00, the wind direction has increased to 156 deg and both sectors operate again in nearly free stream. Accordingly, the rotor-effective wind speeds increase to reach the met-mast reference. Later again, the wind direction varies slightly, leading to partial wake impingements on the right side until, finally (at $\approx$12.00), the wind direction increases further and the horizontal shear becomes almost zero.

Note that the horizontal shear deviates slightly from 0 between 06:00 and 10:30 even though the wind direction is approximately constant around 155 deg. An explanation can be potentially found in the increased turbulence (after sunrise, at around 07:58), that might enhance wake meandering and increase the expansion of the wake. The high turbulence before 02:00 can be attributed to the met-mast being affected by WT2.

This time series very nicely illustrates how the horizontal sector-effective wind speeds and the horizontal shear can be used to understand the instantaneous position of a wake with respect to an affected turbine rotor disk.

Figure 13 reports extended results, showing all available 10-minute values as functions of met-mast wind direction within the range from $\Gamma_{\mathrm{WT2->WT1}} - 45\,\mathrm{deg} = 100$ deg to $\Gamma_{\mathrm{WT2->WT1}} + 45\,\mathrm{deg} = 190$ deg. The waking wind direction from WT2 onto WT1 ($\Gamma_{\mathrm{WT2->WT1}}$) is indicated by a vertical dashed line.

The first subplot from the top shows the rotor-effective wind speed $V_{\mathrm{B}}/V_{\mathrm{ref}}$, non-dimensionalized by the reference speed of the met-mast. Values larger than one can be observed for wind directions close to 100 deg, as the wake of WT2 is affecting the met-mast (see Fig. 2). For wind directions close to 145 deg, lower speeds are observed, caused by the wake of WT2 impinging on WT1. For other wind directions, the speed stays close to one, even though some scatter can be observed.

The central subplot shows the non-dimensional sector-effective wind speeds $V_{\mathrm{S,right}}/V_{\mathrm{ref}}$ and $V_{\mathrm{S,left}}/V_{\mathrm{ref}}$. The small rotor icon shows, using the color code of the subplot, the left (red) and right (blue) sectors. For wind directions between $\approx$ 125 deg

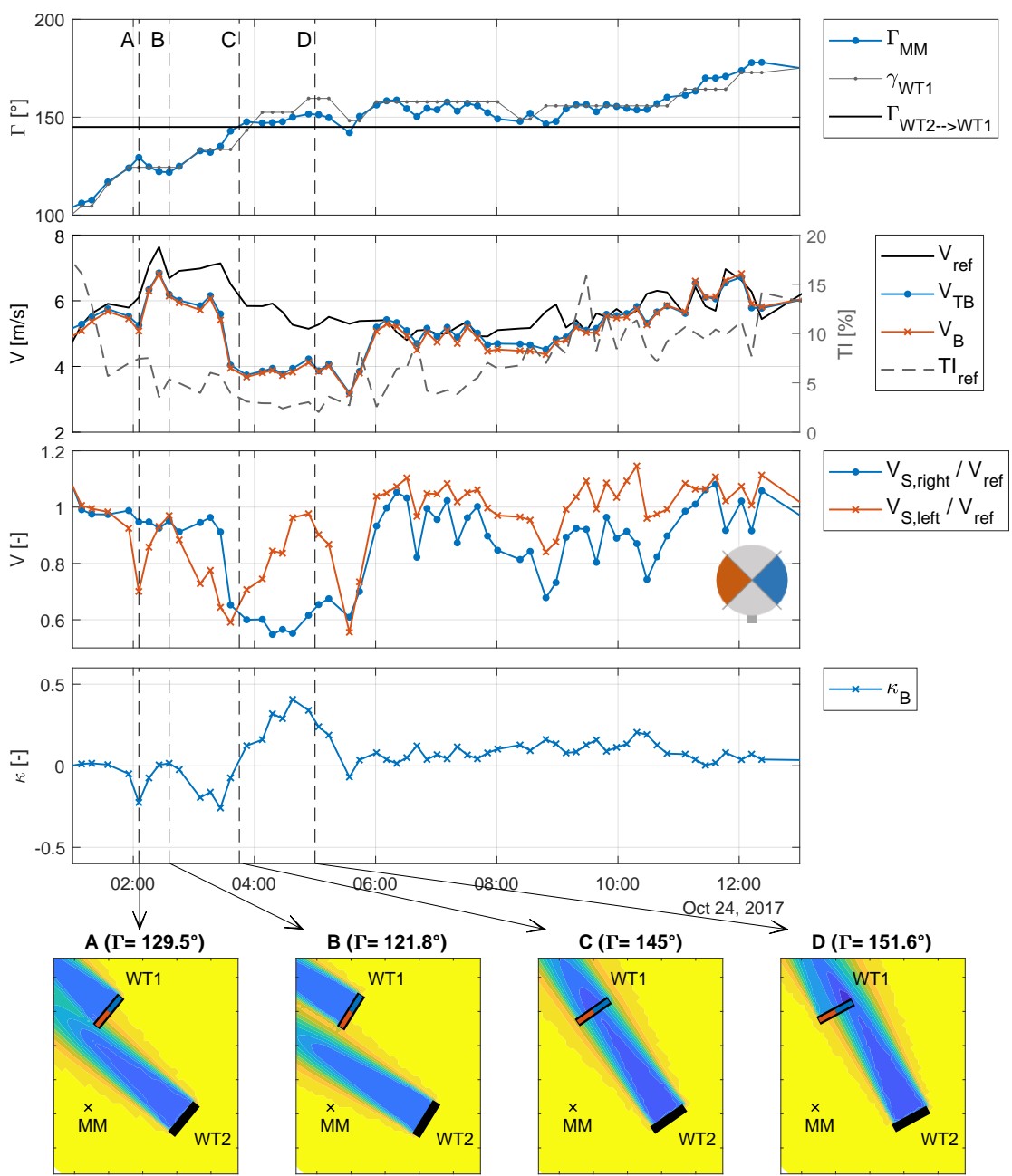

**Figure 12.** Time series characterized by varying wake interference conditions, with met-mast wind direction and turbine yaw orientation (first subplot from the top), reference met-mast wind speed, rotor-effective wind speed estimates and reference turbulence intensity (second subplot), left and right sector-effective speed estimates (third subplot), and horizontal shear estimate (fourth subplot). Lower part of the figure: wake visualizations based on the FLORIS model for different wind directions at time instants A through D.

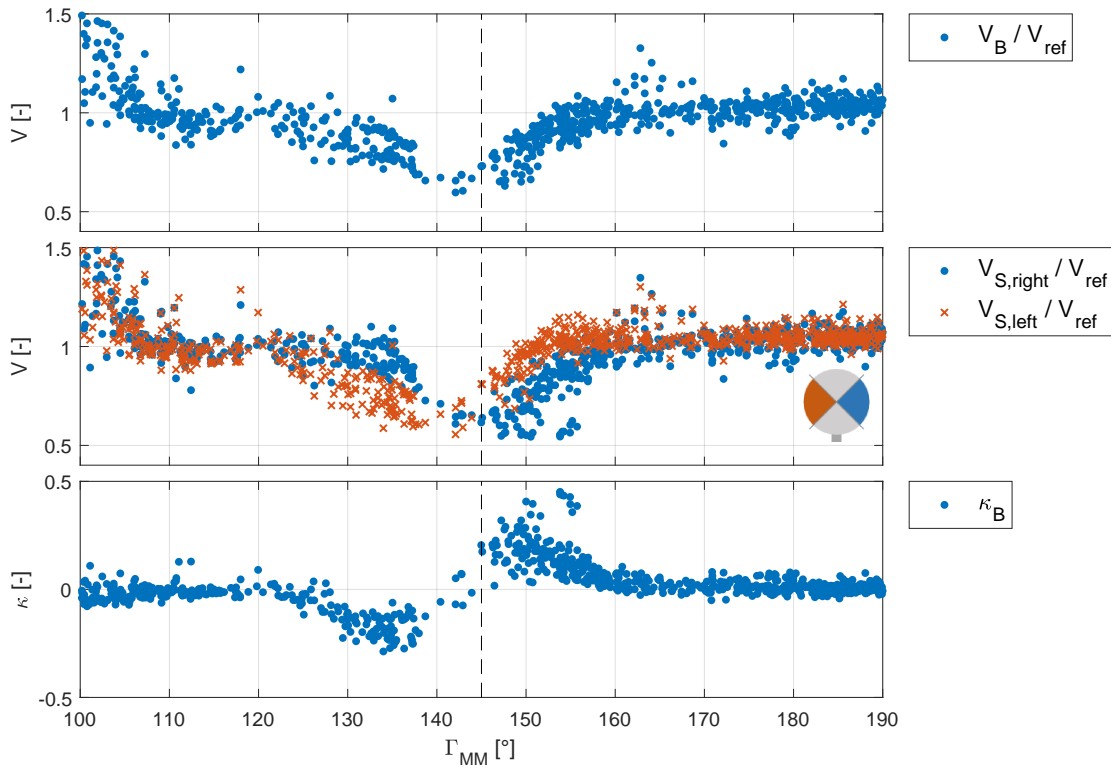

**Figure 13.** Wind speeds and shear at the WT1 rotor disk as functions of wind direction. Top subplot: non-dimensional load-based rotor-effective wind speed. Central subplot: left and right sector-effective wind speeds. Bottom subplot: horizontal shear.

and 140 deg, the local wind speed is smaller in the left sector, indicating that the wake of WT2 is affecting mainly that portion of the rotor disk. Similarly, for wind directions between 145 and about 160 deg, the right sector is affected by the presence of the wake.

The bottom subplot shows the horizontal shear estimate. This quantity is close to zero for all wind conditions, except around the waking direction. Negative values indicate a left-sided wake impingement, while positive values a right-sided one. Note that the scatter observed in the first two subplots, e.g. for wind directions between 160 and 170 deg, seems not to be caused by wake interaction but rather by variations in the reference wind speed, as the horizontal shear is not affected.

For wind directions close to 140 deg, only very few measurement points are available. This suggests that the lower-than-ambient wind speed within the wake of WT2 triggers frequent shutdowns of WT1. The load-based estimator does not operate during turbine shutdowns. Figure 14 shows in 2 deg-wide bins the probability of the WT1 status indicating "no operation". Wind directions were obtained from the met-mast, using all available days without discarding any data point. Indeed, mean direction bins close to $\Gamma_{\mathrm{WT2->WT1}} = 145$ deg support the hypothesis of frequent wake-induced turbine shutdowns. Additionally, Fig. 14 reports a maximum for the bin centered at 141 deg. This, together with the shear shown in Fig. 13, suggests a

small bias in the met-mast wind direction measurement and/or that the wake is not developing exactly along the downstream direction. Indeed, the latter is a phenomenon observed in stable atmospheric conditions when the flow presents a significant vertical shear (Vollmer et al., 2016; Bromm et al., 2018).

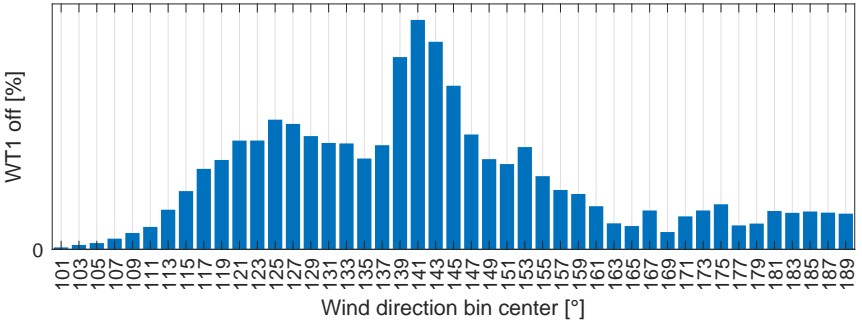

**Figure 14.** Probability of WT2 being in "no operation" state as a function of met-mast wind direction (using 10 Hz measurements of all available days).

These results demonstrate a remarkable ability of the proposed local speed and shear estimates to identify whether and to which extend a downstream turbine operates in the wake of an upstream machine. Note also that the met-mast reference wind direction is just a point measurement at one single height above the ground. In addition, other unknown inflow parameters, such as for example veer, may affect wake development. Therefore, the scatter of some of the data points in Fig. 13 is not necessarily due to inaccuracies of the wind estimator, but might be rather due to the indirect, incomplete and pointwise measurement of the reference wake position.

## 3.9 Effect of turbine misalignment on estimates

As previously mentioned in §2.1, in theory the present method is formulated for turbines aligned with the ambient wind direction. However, in practice this happens only quite rarely, as every turbine in general operates with some degree of misalignment with respect to the incoming wind vector. This is mainly due to two reasons. First, the on-board wind vane(s) may not always provide an exact measurement of the local wind direction. Second, yaw control strategies generally avoid an excessively aggressive tracking of wind direction changes. In fact, a turbine will typically yaw only when its misalignment with the wind has been above a certain threshold for a long-enough duration of time. This is done to limit duty cycle and yaw expenditure, given the very considerable mass of the rotor-nacelle system and the rather modest power capture loss caused by a misalignment of a few degrees.

Since the hypothesis on which the theory is based differs from the situation encountered in practice, it is necessary to show that the typical misalignments of normal turbine operation do not pollute the speed and shear estimates provided by the proposed method. This is achieved here by showing that shears and misalignment are indeed uncorrelated.

To this end, Fig. 15 shows the rotor-effective wind speed as well as the horizontal and vertical shear estimates as functions of the turbine-wind misalignment angle $\Gamma_{\mathrm{rel,WT1}}$. The misalignment is measured using the on-board wind vane. As this instrument may be not always very precise on some turbines, the misalignment angle was also computed by using the met-mast wind direction together with the turbine absolute orientation; however, in the present case no significant difference was observed

5 between these two methods of computing the misalignment angle. The results of the figure only include data points for wind directions between 180 and 337.5 deg, to avoid waked conditions.

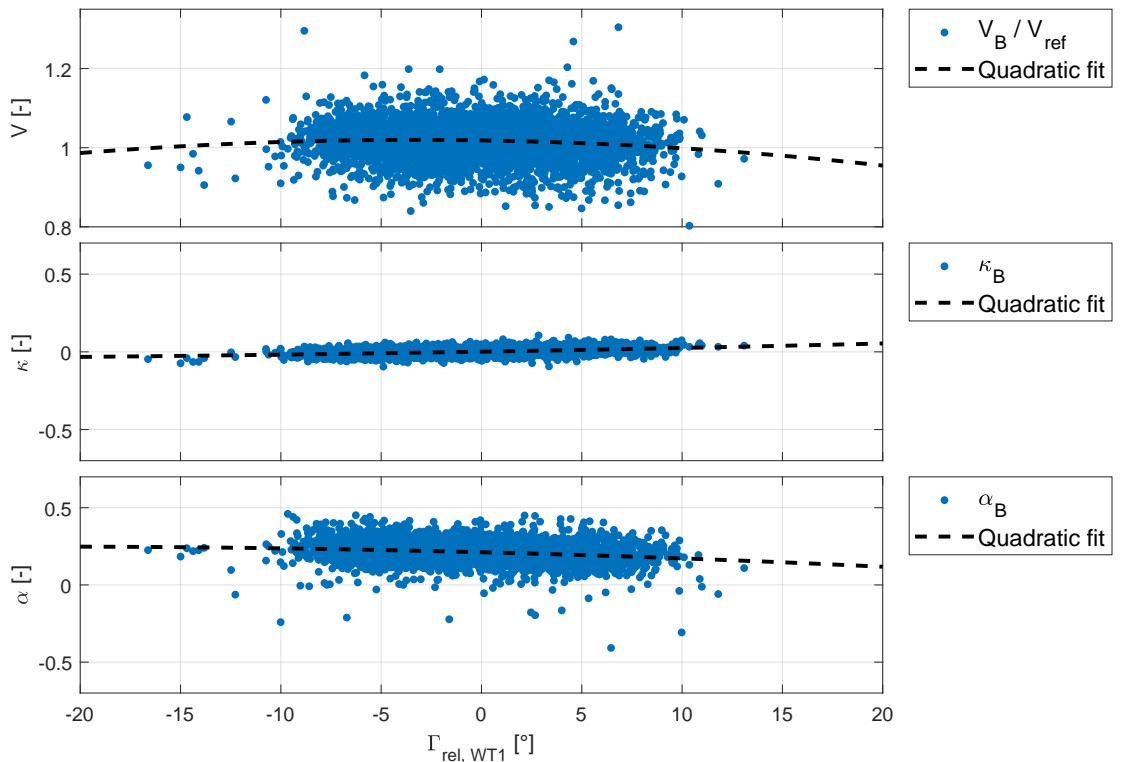

**Figure 15.** From top to bottom: rotor-effective wind speed $V_{\mathrm{B}}$, horizontal shear $\kappa_{\mathrm{B}}$ and vertical shear $\alpha_{\mathrm{B}}$, all plotted as functions of wind turbine misalignment angle $\Gamma_{\mathrm{rel,WT1}}$.

The first subplot reports the non-dimensional rotor-effective wind speed $V_{\mathrm{B}}$. This quantity decreases for increasing misalignment angle, as shown by the second order polynomial fit reported with a dashed line. Such behaviour is completely expected and can be corrected for, if the misalignment is known, by using the cosine law (Gebraad et al., 2015; Fleming et al., 2017;

10 Schreiber et al., 2017).

As shown, the rotor-equivalent wind speed is clearly correlated with misalignment, because the effective speed orthogonal to the rotor plane varies as a function of this angle. However, there is no reason why the vertical and horizontal shears —which are physical characteristics of the inflow— should also exhibit a similar dependency. To verify this fact, the central subplot shows

the horizontal shear estimate $\kappa_B$, which is almost constant with respect to misalignment angle (and also very close to zero). Finally, the bottom subplot shows the vertical shear $\alpha_B$. It appears that both shears have only a very marginal dependency on wind misalignment, as shown by the parabolic best fits reported with dashed lines in the plots. The larger fluctuations of the vertical shear compared to the horizontal one are probably caused by time-varying ambient inflow conditions, as also visible in Fig. 9.

The data shows that the shears are essentially uncorrelated with misalignment. These results demonstrate that the proposed method works without significant errors for turbine-wind misalignment angles up to $\approx \pm 10$ deg.

Larger turbine misalignment angles would be necessary for wake steering control (Fleming et al., 2017), where the rotor is intentionally pointed away from the wind to deflect the wake laterally. The performance of the proposed method could not be tested in such conditions within the present research, as no large misalignment angles were present in the available data set. However, even in that case, the procedure illustrated here could be used for pragmatically correcting possible errors caused by misalignment. In fact, by plotting shears as functions of misalignment angle, a best-fit correction function could be readily derived and used for correcting the estimates, if necessary.

## 4    Conclusions

A method to estimate the local wind speeds over sectors of the rotor disk has been tested on a 3.5 MW wind turbine. Results have been compared to reference values obtained with a nearby met-mast. For some wind directions, the sensing turbine is waked by a second machine. This feature of the test site has been exploited to test the ability of the proposed local wind sensing technique to detect wake impingement.

The wind sensing method has been previously studied and evaluated in simulations and scaled experiments. The present work has presented a first full-scale demonstration. Based on the field test results shown herein, the following conclusions can be drawn:

- A rotor-effective wind speed can be estimated from blade out-of-plane bending moments, with a quality that is nearly indistinguishable from the well known torque-balance method.

- The vertical wind shear estimated from out-of-plane bending moments correlates very well with the met-mast reference. The best results were obtained when the mast is directly upstream of the turbine. This suggests that some of the scatter in the results might be due to a lack of knowledge of the exact ground truth, rather than to a lack of accuracy of the proposed method.

- The vertical shear measured by the met-mast up to hub height differs from the shear measured over the whole rotor disk. This is likely a feature of the flow, and not of the method tested here.

- The local wind speeds estimated on two lateral sectors of the rotor disk show the clear fingerprint of an impinging wake shed by a neighbouring turbine. By looking at the two sectors, one can distinguish left, right or full wake overlaps.

- Simple and very practical techniques can be used to correct for various sources of error, including not perfectly calibrated load or azimuth sensors, as well as model approximations.

The present load-based wind estimation method provides for a remarkably simple and effective opportunity to estimate atmospheric inflow conditions on operating turbines. The method is based on readily-available quantities that can be easily
5   computed from a standard model of a wind turbine, and does not need to be trained from extensive data sets. The on-board implementation uses pre-computed look-up-tables. and hence has a negligible computational cost. When load sensors are already installed on a turbine, for example for load-reducing control, this novel wind sensing capability is simply obtained as a software upgrade. Wind sensing opens up a number of opportunities that can profit from a better knowledge of the inflow, including turbine and wind farm control, lifetime consumption estimation, predictive maintenance and forecasting, among
10  others.

*Code and data availability.*  The operational data and turbine model used in this research are the property of eno energy systems GmbH. An implementation of the estimator can be obtained by contacting the authors.

## Nomenclature

| | |
|---|---|
| BEM | Blade Element Momentum |
| LUT | Look-up-table |
| MM | Met-mast or meteorological mast |
| WT1 | Wind turbine 1 (sensing turbine) |
| WT2 | Wind turbine 2 |
| | |
| $a$ | Linear best fit constant ($y = ax + b$) |
| $A$ | Rotor disk area |
| $A_{\mathrm{S}}$ | Sector area |
| $b$ | Linear best fit constant ($y = ax + b$) |
| $C_{\mathrm{m}}$ | Cone coefficient |
| $C_{\mathrm{p}}$ | Power coefficient |
| $c$ | Speed estimate scaling factor |
| $D$ | Rotor diameter |
| $J$ | Total rotational inertia |
| $m_i$ | Blade root out-of-plane bending moment of blade $i$ |
| $m_{i,\mathrm{meas}}$ | Measured blade root out-of-plane bending moment of blade $i$ |
| $N$ | Number of measurements |

| | | |
|---|---|---|
| | $q$ | Dynamic pressure |
| | $R$ | Rotor radius or Pearson correlation coefficient |
| | $s$ | Load scaling factor |
| | $s_{\mathrm{MM->WT1}}$ | Downstream distance between met-mast and wind turbine WT1 |
| 5 | $T_{\mathrm{aero}}$ | Aerodynamic torque |
| | $T_{\mathrm{meas}}$ | Measured torque |
| | $\mathrm{TI_{ref}}$ | Met-mast measured reference turbulence intensity |
| | $V$ | Wind speed |
| | $V_{\mathrm{B}}$ | Blade-load estimated rotor-effective wind speed |
| 10 | $V_{\mathrm{H}}$ | Wind speed at hub height |
| | $V_i$ | Blade-effective wind speed estimate of blade $i$ |
| | $V_{\mathrm{MM},i}$ | Met-mast measured wind speed at height $i$ |
| | $V_{\mathrm{PL}}$ | Power law inflow profile |
| | $V_{\mathrm{ref}}$ | Met-mast measured reference wind speed of inflow profile |
| 15 | $V_{\mathrm{S}}$ | Sector-effective wind speed |
| | $V_{\mathrm{S,left/right/up/down}}$ | Load-based estimation of left/right/up/down sector |
| | $V_{\mathrm{TB}}$ | Torque-balance estimated rotor-effective wind speed |
| | $y$ | Lateral position |
| | $z$ | Height above ground |
| 20 | $z_{\mathrm{H}}$ | Hub height |
| | $z_{\mathrm{MM},i}$ | Installation height of sensor $i$ on met-mast |
| | | |
| | $\alpha$ | Vertical shear exponent |
| | $\alpha_{\mathrm{B}}$ | Load-based estimated vertical shear exponent |
| 25 | $\alpha_{\mathrm{lower,B}}$ | Load-based estimated vertical shear exponent on lower half of rotor disk |
| | $\alpha_{\mathrm{MM}}$ | Met-mast-measured vertical shear exponent |
| | $\beta$ | Blade pitch angle |
| | $\gamma$ | Turbine yaw orientation (clockwise from due North) |
| | $\Gamma$ | Wind direction (clockwise from due North) |
| 30 | $\Gamma_{\mathrm{A->B}}$ | Direction of wind blowing from point A to B (clockwise from due North) |
| | $\Gamma_{\mathrm{MM}}$ | Wind direction at met-mast |
| | $\Gamma_{\mathrm{rel,WT1}}$ | Relative wind direction at nacelle of WT1 |
| | $\Delta t$ | Time delay between measurement at met-mast and turbine |
| | $\epsilon_{\mathrm{RMS}}$ | Root mean squared error |
| 35 | $\kappa$ | Horizontal shear coefficient |

| | | |
|---|---|---|
| $\kappa_{\mathrm{B}}$ | | Load-based estimated horizontal shear coefficient |
| $\lambda$ | | Tip speed ratio |
| $\rho$ | | Air density |
| $\rho_{\mathrm{ref}}$ | | Reference air density |
| 5 | $\psi$ | Blade azimuth position |
| | $\psi_{\mathrm{a}}$ | Blade azimuth position, beginning of sector |
| | $\psi_{\mathrm{b}}$ | Blade azimuth position, end of sector |
| | $\psi_{\mathrm{bias}}$ | Blade azimuth measurement offset |
| | $\psi_{\mathrm{corr}}$ | Corrected azimuth measurement |
| 10 | $\Omega$ | Rotor speed |
| | $\dot{\Omega}$ | Rotor acceleration |

*Author contributions.* JS conducted the main research work and prepared a first draft of the manuscript. CLB developed the core idea of load-based wind sensing, supervised the research and contributed to the writing of the paper. MB assisted in the measurement post-processing and analysis. All authors provided important input to this research work through discussions, feedback and by improving the manuscript.

*Competing interests.* The authors declare that they have no conflict of interest.

*Acknowledgements.* The authors express their gratitude to Stefan Bockholt and Alexander Gerds of eno energy systems GmbH, who granted access to the measurement data and turbine model, and to Marijn van Dooren, Anantha Sekar and Martin Kühn of ForWind Oldenburg, who shared insight on the data. This work has been supported by the CompactWind II project (FKZ: 0325492G), which receives funding from the German Federal Ministry for Economic Affairs and Energy (BMWi).

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
