# Peer review of "Field testing of a local wind inflow estimator and wake detector"

_Wind Energy Science, 2020_

## Short Comment (SC1) · 23 Mar 2020

Dear authors,

You mention on p.10 lines 11-16 the method for eradicating the bias in the azimuth readings, due to the sensors or blade dynamics. Could you please comment why you would expect the bias from ignoring blade dynamics to be a constant, independent of wind speed? It might be a constant for the sensor bias over the period investigated (it might drift with time), but it is hard to see this to be the case for blade dynamics or are the first 7 days representative for the entire measurement period? It would be interesting to plot $\phi_{bias}$ as a function of time or wind speed. It should be simple to incorporate a variable $\phi_{bias}$ in your method and it might be necessary for long-term

applications.

Thanks
* * *

---

## Referee Comment (RC1) · Anonymous Referee #1 · 15 Apr 2020

**Journal:** WES
**MS No.:** wes-2020-48
**MS Type:** Research articles
**Submission Date**
**Date Due** 15 Mars 2020
**Title:** "Field testing of a local wind inflow estimator and wake detector"
**Author(s):** Johannes Schreiber, Carlo L. Bottasso, and Marta Bertelè

**General comments:**

This paper is in the continuity of other studies on the same subject. The objective of this work and the previous ones is to predict the mean wind state inflows (shears and misalignments) from wind turbine loads. Previous works were dedicated on the validation of the method using an aeroelastic simulations, LES simulations and scaled wind tunnel tests .The present work extend the validation using field tests.

Having the knowledge of the flow affecting the entire rotor including its impact on the production for all wind directions/conditions is indeed not trivial with today's sensors that are generally limited spatially (a point measurements with nacelle-mounted wind lidar, some points of the vertical wind profile from a met-mast ...), limited to certain wind directions (met-mast location, scanning lidar position etc …) and limited in time (met-mast/scanning lidar are generally installed for the certain amount of time). Extract information of the wind inflow from remote sensors on the rotor blades, that can be included in the monitoring set of data (SCADA data), is therefore a very attractive solution.

The use of blade out-of-plane bending moment is demonstrated here to be an interesting quantity for wind inflow analysis. However, a sensor characterization is generally based on measurements redundancy. For this sensor this means a perfect knowledge of the spatio-temporal inflow, which is only partially available from this field test. Also, sensors need to be characterized dynamically which is a quite complex task from field measurements. In antoher hand, reproducing all the physics in wind tunnel or in simulations is still a challenging task and this first application of the method on field test is a real interesting feedback to perform further accuracy evaluations that could be completed off-line in controlled environment using wind tunnel tests, CFD or aero-elastic simulations.

I clearly recommend this paper for publication with however some corrections or more details regarding these tow points:

Mean shear trends can be retrieved after however some online calibrations:
- calibration coefficient introduced to compensate the mismatched of the bending moments on the two blades
- calibration coefficient introduced to match the rotor effective speed VTB with the velocity estimated from the bending moment measurement.
- azimuth mean biais of 11.4° is removed (using the average over 7 days).

The origin of the mismatch is not always completely evaluated. If the source of the error dependent on the rotor operation, this will limits the validation of the method to this specific rotor operating point. It would reinforce the strength of the paper to look in more details at the origin of these mismatch.

For that purpose, it would be interesting to have more informations on the available sensors (type, accuracy, calibration procedure ) such as the azimuthal sensor, the sensor available on the mast, the initial strain gauge calibration … This is particularly important to help to discriminate the error from the model to the measurement and thus to have more inside on the origin of some errors/biais found by the authors.

**DETAILED QUESTIONS:**

**Q1:** Measurements used in the paper are 10min averaged data. However, it is particularly interesting to have an estimation of the wind fluctuations at the rotor location for blade load monitoring/alleviation for instance. The highest time resolution for this method/sensor  is linked to the strain gauge sensor cut-off frequency, to the structural dynamic response of the blade bending moment, but also to the rotation speed of the rotor. The rotation speed of the rotor is varying with wind inflow according to the control of the turbine, so that the developed sensor has a varying sampling rate. Have you estimated the sampling rate variations ? Does it impact the wind estimation ? Do you have an estimate of the minimal/maximum time resolution for a given azimuth position (phase measurements) ?

**Q2: p3L22 " (…) former yields a rotor-effective wind speed (i.e., an average quantity over the entire rotor disk), the latter is used to sample the local wind speed at the azimuth position occupied by a blade."**

With strain gauge sensors only located at the root for root bending moment measurements (with the out-of-plane forces assumed to be homogeneously distributed along the blade), the estimation of the associated wind condition is necessarily averaged along the blade. This method is therefore local in azimuth, but not along the blade. I think this is an important information to be emphasized as it is more complex to install strain gauge sensors along the blade (for more local estimation) than only at the blade root location.

**Q3 P6L22: "All measurements are sampled at 10Hz".**
Why not using the 10Hz data, why only the 10 min average ?

**Q4: p6L25: "the relative difference between the two blades can't be related to a miscalibration of sensors …"**
why not a small pitch offset beween blades ?
The cross-checking of the the load calibration is given through a comparison between rotor-effective wind speed and the wind from blade loads. However, no information is available on the initial calibration of the strain gauges, which is an important point to evaluate the accuracy of these measurements and thus to discriminate between an error in measurement and a lack in the model development or other source of errors.

**Q5 p8L25: "including the cases where the blade is partially or fully stalled"**
CD and CL are inputs given to the aero-elastic modeling. How these cases are treated ? Is this a LUT of measured CL/CD or a Xfoil simulation ? Or aerodyn from fast ? Or CFD computations … ?

**Q6 p8L30: "A direct comparison between VTB and VB reveals that the latter provides … are scaled by a factor of c=0.928"**

Why the model used to compute the aerodynamic coefficients is suspected to be the source of

error ? Is the model used limited ? Are the operating AoA in the stall region ?

Why not a misalignment bias of the rotor or difference of pitch angles between blades during installation ?

**Q7 p10: "possible bias in the measurement of the azimuthal position of the rotor" or "no blade dynamics included in the model"**

How the azimuth is measured, is the 11.4° in the error range of the sensor ?

Why you didn't include the blade dynamic model ? This would have been interesting to cross-check your hypothesis and discriminate between a sensor error or a modeling error.

**Q8 p15 figure 10:**

Is it possible to have the floris pictures between instant C and instant 5:00, where there is a peak increase of velocity $V_{s,left}$ ?

It seems to me that the rotor orientation hasn't changed much relatively to the instant C (gama is constant ~145° ) while the $V_{s,left}$ peak is quite significant and the $V_{s,right}$ remain constant (waked condition). This dissymmetry in the wind estimation (and therefore in load bending moments) is quite strange if the wind orientation hasn't changed. Maybe an errorbar in the measurement of the wind orientation may help ?

**Q9: p15 figure 10**

Another point that is remarkable is instant ~9:00. While the wind direction is back to the level found after instant C (~149°), the deficit is not as high and the dissymmetry between $V_{s,left}$ and $V_{s,right}$ is again present. I suspect a too fast wind direction variation for the wake to develop. In another word, apart from errors in the method, is the wind unsteadiness can be suspected.

Standard deviation of the wind direction may help to go a bit further in the analysis.

I understand that without reference this is difficult to explain, however this high sensitivity to the time duration within a wind orientation is certainly to be estimated off-line with a dynamic calibration of the sensor method in future work. It should be at minimum reported or commented in the present paper.

**Q10 (figure 10):** the coefficient k is interesting but not commented, why is that ?

The passage from a positive shear to a negative shear, the level of the shear at 5:00 compared to 9:00 etc ...

**Q11 P17L5: "very few measurement points are available" induces "frequent shutdows of WT1"**

Can you be clearer ? I don't understand this logic: even if the wind turbine is stopped you should have bending moments measurement points ?

**Q12 p17L10: "Fig. 11, suggests a small bias in the met-mast wind direction measurement and/ or that the wake is not developing exactly along the downstream direction."**

Also suggested by figure 10 with the dissymmetry between $V_{s,left}$, $V_{s,right}$ ?

**Q13 P17L16: "the scatter ..."**

It can also be attributed to the level of the atmospheric turbulence in the inflow, a comparison from std from met-mast and std of $V_{s,right}$ / $V_{s,left}$ may help to assess this point ?

**Q14 p18L25: "The larger fluctuations of the vertical shear compared to the horizontal one are**

**probably caused by varying ambient inflow conditions.”**

Depending on the mast instrumentation (sonic or vanes), this point can be assessed by the evaluation of the atmospheric stability and thus possible additional velocity fluctuations in the vertical direction.

**Q15:**
**P19-20: “This indicates that some of the scatter ...proposed method”**
**P20L4: “Clearly, this is simply a feature of the flow, and not of the method tested here.”**

These sentenses are very affirmative while there was no clear demonstration on that purpose.

Clearly tendencies agree well with what is expected and the method gives interesting results. However, additional measurement points are needed to have an effective measure of the method accuracy in space (more points on the mast in the vertical direction, maybe a mast in the horizontal direction, and some topological analysis of the terrain …).

**Q16 p19L4: “(…) waked by a second machine. This feature of the test site has been exploited for demonstrating the ability of the proposed local wind sensing technique to detect wake impingement.”**

The measurements available on field test site is not able to perform a direct validation of the method, which would consist on a direct comparison between a full spatio-temporal description of the wind inflow (at least a 2D plan) with the estimated one. The demonstration is rather based on analysis from partially available measurements (mast, SCADA, azimuth, …) completed with wake estimation from FLORIS. More specifically, there is no way to validate the horizontal shear (wake) with inflow measurements (only one point). Tendencies are clearly coherent to what we would expect, but a precise evaluation of the method accuracy (in time and space) is not feasible.

The term “demonstrating” is therefore a bit strong here, especially for the wake detection.

**MINOR CORRECTIONS:**

**C1:** In equation 1a, V should be replaced by VTB and in equation 1b, V should be replaced by Vi

**C2:** Usual conventions for wind roses representations are: North corresponds to *0°*/360°, East to 90°, South to 180° and West to 270°. In figure 2, 0°/360° corresponds to South.

---

## Referee Comment (RC2) · Anonymous Referee #2 · 18 Apr 2020

Thank you for this paper. I apologize up front that due to school closures and work hour impacts, this will be a brief review. That said, the paper is of high quality such that I have very little in the way of criticism. The paper follows a set of earlier papers (described in the introduction) which develop the methods tested in this paper, and evaluate it in aero-elastic, LES and wind tunnel testing. The current paper tests the estimation approaches on a full-scale test site. The results are completely convincing. The presence of the nearby met mast offers a very good validation to compare estimation of speed, shear and wake position and the analysis is clear and direct to follow, the conclusions well-justified by the presented figures. Finally, the introduction and literature are well covered, and the paper put well in the context of the broader research areas which could utilize estimation like this. I checked the equations and didn't notice

any obvious errors. Recommend accepting.

Small comments:

1) Is the cone coefficient a standard value, or an innovation of an earlier paper in this series?

2) Section 3.6: "Using again the first 7 days of measurements, the azimuth bias was identified as $\psi$bias = 11.4 âŮę . 15 In the remainder of this work, the sector-effective wind speeds and the two shears are computed using the corrected azimuth signal $\psi$corr = $\psi$ + $\psi$bias."

This was interesting, as it reminds me off the offset value one might compute in the design of standard IPC controllers for 1P or 2P decoupling. Is this the same value?

---

## Referee Comment (RC3) · Anonymous Referee #3 · 20 Apr 2020

The manuscript entitled "Field testing of a local wind inflow estimator and wake detector" deals with the full-scale experimental validation of estimator concepts based on the use of the rotor as a wind sensor. The methods are based on the processing of the blade load fluctuations, and particularly the blade out-of-plane bending moments. Since 2010, the research team lead by Bottasso developed, improved and validated the concept of using the rotor as a wind sensor and the present paper is in line with this continuous process. It reaches a new step, by performing the demonstration and partial validation of the concept at full scale, on utility-scale wind turbines. The main challenges are then to obtain statistically converged, reliable and exploitable results when the boundary conditions of the experiments are non-controllable and partially known (onsite environmental and atmospheric conditions) and when the propotype

is not specifically designed and equipped for R&D purpose experiments (utility-scale wind turbine). These aspects lead to the need for an extensive preparation of the database by using massive data pre-processing (ad-hoc calibrations and corrections, sample rejections, filtering, classification, etc.). In the present paper, these unavoidable pre-processing steps, as well as the actual data processing steps, are well argued, described and illustrated. The obtained results are on general, well explained and prove the feasibility of the "WT as a wind sensor" concept. On the other hand, a lack of information on the experimental set-up and on the site description affect sometimes the reliability of the results interpretation, leading the authors to use too frequently "likely", would", "seems to", could be due to". Mainly, a better knowledge of the site properties (terrain and micrometeorology) can help in some interpretations. This can be provided a posteriori using geographical and meteorological databases and it is essential to add them to the manuscript.

Major comments: - A thorough description of the experimental set-up must be added: measurement device (anemometers, strain gages, etc.) descriptions (type, brand, accuracy, cut-off frequency, etc.)  - A thorough description of the site properties must be added: type of terrain surrounding the site (type of vegetation, associated roughness length), atmospheric boundary layer properties (wind rose, averaged power law and turbulence intensity at hub height for the studied wind directions, thermal stability encountered during the selected periods, etc.). If not findable by the measurement campaign itself, meteorological information can be extracted from global meteorology reanalysis database as MERRA2 or ERA5. - §3.3 Reference inflow & Figure 3 : it is written that the wind speed is measured at three different heights on the met-mast but two of them are located at 2m of each other. Therefore, one cannot consider that one has three distinct values to assess the power law exponent, but only two. What is the consequence on the accuracy of the obtained power law exponent? - Figures 7 and 8: The obtained values for the power law exponent (mainly between 0.2 and 0.4) are particularly high for such an open-field terrain, as it seems to be on the satellite picture. These values are usually encountered on rough to very rough terrains (forest

or city). Again, a better description of the terrain fetch and of the local atmospheric boundary layer properties would help to justify the results reliability. - Page 12, lines 9-10: "This difference could possibly be caused by a non-ideal power law inflow profile, leading to a biased met-mast reference shear, although a definitive explanation of this mismatch could not be reached with the present data set.". I would recommend to make a sensitivity analysis on the power law exponent to the number and position of the used anemometers - Pages 12-13: "Considering that all wind directions are for un-waked met-mast and turbine, these results suggest the presence of a spatial shear variation, probably caused by the local vegetation." Again, a better description of the terrain fetch and of the local atmospheric boundary layer properties would help to justify this assumption. - - It is written on page 8, lines 4-5, "Measurements taken during yawing manoeuvres were also discarded, as additional induced loads can pollute the estimates". On the other hand, on Figures 6 and 10, the wind direction progressively changes from 240° to 200° during 6 hours, and from 100° to 175] in 12 hours, respectively. Yaw manoeuvres should appear during these periods. It sounds in opposition of the first statement. Could you please add the wind turbine orientation time series to these plots and explain how you did the data analysis during these periods? - Figure 11 : would it be possible to classify the results considering the incoming wind speed category (and so the wind turbine operating point). One could expect that the wake is more or less intense, depending on the wind turbine operating point and that the wake detector is more or less efficient. - - Page 17, lines 10-12: "the wake is not developing exactly along the downstream direction. Indeed, the latter is a well-known phenomenon observed in vertically sheared flow (Vollmer et al., 2016)." Yes, it is true for yawed wind turbines, or for un-yawed ones in very stable atmospheric conditions but cannot be considered as a universal explanation for the bias in the present results. - Page 18, lines 24-25: "The larger fluctuations of the vertical shear compared to the horizontal one are probably caused by varying ambient inflow conditions." It is not clear what this sentence means exactly. Could you elaborate more on these "varying ambient inflow conditions"? Again, a better knowledge of the local atmospheric boundary

layer properties can help to justify some results. - Conclusions: some conclusions are not new (i.e. "rotor-effective wind speed can be estimated from blade out-of-plane bending moments, with a quality that is nearly indistinguishable from the well-known torque-balance method"), since already drawn by previous papers from the same research team. What is new is to make the full-scale demonstration/validation of these different concepts.

Minor comments: -Page 3, line 17: remove A in the q formula - Page 4, line 3-4: "A rotor-effective wind speed can also be obtained from the blade-effective ones by simple averaging over all (three) blades". One expects that the dynamics of the rotor-effective speed is quite low (cut-off frequency linked to the rotor diameter, whereas the dynamics of the blade-effective ones must be higher. Do you get the right rotor-effective speed dynamics by averaging the three blade-effective wind speeds? - page 4, line 17-18 "he smaller inertia and high damping of this degree of freedom makes this more sophisticated approach superfluous": Please add a reference to prove this statement. - Page 5, figure 1: the reference framework (x,y,z) is not direct. Considering the naming convention in the downstream viewing direction, one assumes that x is in the downstream direction too. Then y, should be oriented on the left - Page 9, lines 7-8: add this information into the experimental set-up description - Figures 4& 5: should be written in the captions that it is after correction

---

## Author Comment (AC1) · 20 May 2020

Dear reviewers, dear Alexander Meyer Forsting,

thank you for your constructive and useful input. Please find enclosed a detailed point-by-point reply to all of your comments. We are also enclosing a revised manuscript, with highlighted changes with respect to the previous version of the paper.

Thanks again, kind regards Carlo L. Bottasso (on behalf of the authors)

Please also note the supplement to this comment:
https://www.wind-energ-sci-discuss.net/wes-2020-48/wes-2020-48-AC1-supplement.pdf

[Figure]

**Supplement:**

**Reply to reviewers**

We thank the reviewers who, with their detailed analyses and constructive inputs, have helped improve the quality of this paper. A list of point-by-point replies to their comments is reported in the following, and a revised version of the manuscript is attached with highlighted changes.
In addition, we have taken the opportunity for several minor improvements to the text to increase clarity or form.

Best regards,
The authors

**Contents**

**Short comment 1 (SC1)**

**Comment**

Dear authors,
You mention on p.10 lines 11-16 the method for eradicating the bias in the azimuth readings, due to the sensors or blade dynamics. Could you please comment why you would expect the bias from ignoring blade dynamics to be a constant, independent of wind speed? It might be a constant for the sensor bias over the period investigated (it might drift with time), but it is hard to see this to be the case for blade dynamics or are the first 7 days representative for the entire measurement period? It would be interesting to plot $\psi_{bias}$ as a function of time or wind speed. It should be simple to incorporate a variable $\psi_{bias}$ in your method and it might be necessary for long-term applications.
Thanks

Authors:
Thank you for your comment. Indeed, the azimuth bias can be affected by blade dynamics (and therefore it might also depend on wind or rotor speed). The figures below show the azimuth bias identification for varying wind and rotor speeds using the complete dataset (in 1 m/s and 1 RPM bins respectively, N showing the number of samples in each bin).

[Figure]

As suggested in the comment, a rotor speed or wind speed dependency can be implemented to take such dynamic effects into account, for example using a one dimensional look-up-table. For a long term field implementation such an approach is indeed reasonable and should probably be adopted. However, in this work we preferred the use of a constant average value, mainly to simplify the analysis and discussion, but also because a variable bias does not significantly improve the results. Also, the azimuth bias might change over time, an effect that we were not able to observe with sufficient accuracy because only 41 days were included in the dataset. In fact, for the first 7 days we identified a bias of 14.8 deg, while a bias of 16.3 deg was found for the last 7 days. However, this variation might also have been caused by a different wind/rotor speed distribution, so that we opted even in this case for an average value.
We have updated the manuscript to explain the points above.
Finally, we have found that the manuscript erroneously reported a bias of 11.4 deg for the first 7 days (instead of 14.8 deg). This error has been corrected in all analyses, with no significant effect on the results.

**General comments**

This paper is in the continuity of other studies on the same subject. The objective of this work and the previous ones is to predict the mean wind state inflows (shears and misalignments) from wind turbine loads. Previous works were dedicated on the validation of the method using an aeroelastic simulations, LES simulations and scaled wind tunnel tests .The present work extend the validation using field tests.

Having the knowledge of the flow affecting the entire rotor including its impact on the production for all wind directions/conditions is indeed not trivial with today's sensors that are generally limited spatially (a point measurements with nacelle-mounted wind lidar, some points of the vertical wind profile from a met-mast ...), limited to certain wind directions (met-mast location, scanning lidar position etc …) and limited in time (met-mast/scanning lidar are generally installed for the certain amount of time). Extract information of the wind inflow from remote sensors on the rotor blades, that can be included in the monitoring set of data (SCADA data), is therefore a very attractive solution.

The use of blade out-of-plane bending moment is demonstrated here to be an interesting quantity for wind inflow analysis. However, a sensor characterization is generally based on measurements redundancy. For this sensor this means a perfect knowledge of the spatio-temporal inflow, which is only partially available from this field test. Also, sensors need to be characterized dynamically which is a quite complex task from field measurements. In antoher hand, reproducing all the physics in wind tunnel or in simulations is still a challenging task and this first application of the method on field test is a real interesting feedback to perform further accuracy evaluations that could be completed off-line in controlled environment using wind tunnel tests, CFD or aero-elastic simulations.

I clearly recommend this paper for publication …

**Authors:**
Thank you for your positive review and helpful comments.

… with however some corrections or more details regarding these tow points:
Mean shear trends can be retrieved after however some online calibrations:
- calibration coefficient introduced to compensate the mismatched of the bending moments on the two blades
- calibration coefficient introduced to match the rotor effective speed VTB with the velocity estimated from the bending moment measurement.
- azimuth mean biais of 11.4° is removed (using the average over 7 days).

The origin of the mismatch is not always completely evaluated. If the source of the error dependent on the rotor operation, this will limits the validation of the method to this specific rotor operating point. It would reinforce the strength of the paper to look in more details at the origin of these mismatch.

**Authors:**
Errors affecting the sensors of different machines could clearly be of different origin and entity. However, the calibration procedures that were used in this paper are general, and could be applied to each different turbine to calibrate its sensors. It should be realized that these calibration procedures are independent from the cause of the miscalibration, which is an important advantage. We do not see the limit mentioned by the reviewer related to the rotor operating point. For example,

the calibration of the load sensors simply enforces the same long-term averages among the various blades, so there is no dependency on the operating point as indeed many different points are used. The calibration of the azimuth bias was also based on a 7-day average (i.e., many different operating points), although a more sophisticated calibration that depends on the rotor speed could be used (see reply above to SC1).
We have improved the descriptions of the calibration procedures, and especially expanded the one of the rotor blades and of the azimuth bias.

For that purpose, it would be interesting to have more informations on the available sensors (type, accuracy, calibration procedure) such as the azimuthal sensor, the sensor available on the mast, the initial strain gauge calibration … This is particularly important to help to discriminate the error from the model to the measurement and thus to have more inside on the origin of some errors/biais found by the authors.

Authors:
Additional technical specifications of the sensors have now been included in the manuscript. Even more information on the origin of errors and biases would certainly be valuable. However, such information is often unavailable in practical field applications, as also stated by Bromm 2017: *"All sensors are subject to measurement uncertainties, and their perfect mounting and alignment cannot be automatically assumed in a field environment"*. As the sensors can be miscalibrated for many different reasons, this work explicitly proposes methods to correct them in a pragmatic cause-independent way.

**Detailed questions**

**Q1:** Measurements used in the paper are 10min averaged data. However, it is particularly interesting to have an estimation of the wind fluctuations at the rotor location for blade load monitoring/alleviation for instance. The highest time resolution for this method/sensor is linked to the strain gauge sensor cut-off frequency, to the structural dynamic response of the blade bending moment, but also to the rotation speed of the rotor. The rotation speed of the rotor is varying with wind inflow according to the control of the turbine, so that the developed sensor has a varying sampling rate. Have you estimated the sampling rate variations? Does it impact the wind estimation? Do you have an estimate of the minimal/maximum time resolution for a given azimuth position (phase measurements)?

Authors:
We have included a new section "3.3 Estimator update frequency" to address the reviewer's question. In short, we show how even the slowest update frequency (occurring at low rotor speeds) is still capable of resolving the main fluctuations of the inflow.

**Q2: p3L22 " (…) former yields a rotor-effective wind speed (i.e., an average quantity over the entire rotor disk), the latter is used to sample the local wind speed at the azimuth position occupied by a blade."**
With strain gauge sensors only located at the root for root bending moment measurements (with the out-of-plane forces assumed to be homogeneously distributed along the blade), the estimation of the associated wind condition is necessarily averaged along the blade. This method is therefore local in azimuth, but not along the blade. I think this is an important information to be emphasized

as it is more complex to install strain gauge sensors along the blade (for more local estimation) than only at the blade root location.

Authors:
We have updated the description of the formulation to include the point noted by the reviewer.

**Q3 P6L22: "All measurements are sampled at 10Hz".**
Why not using the 10Hz data, why only the 10 min average ?

Authors:
The choice of 10 min averages is now explained in more detail. In addition, an analysis of the higher frequency signals has been included and is shown in Section 3.7.

**Q4: p6L25: "the relative difference between the two blades can't be related to a miscalibration of sensors …"**
Why not a small pitch offset beween blades? The cross-checking of the the load calibration is given through a comparison between rotoreffective wind speed and the wind from blade loads. However, no information is available on the initial calibration of the strain gauges, which is an important point to evaluate the accuracy of these measurements and thus to discriminate between an error in measurement and a lack in the model development or other source of errors.

Authors:
We agree, the difference might be caused also by a small pitch offset. Possible causes of the miscalibration have now been included in the manuscript.
The initial calibration of the strain gauges is not known to the authors. We agree that a discrimination among the possible sources of error is of potential interest, but such an analysis seems not to be possible with the present data set. In addition, it should be noted that a main goal of this paper is the realistic demonstration of the method – and in a realistic scenario detailed information and root causes of all possible errors are also often unknown (Bromm, 2017).

**Q5 p8L25: "including the cases where the blade is partially or fully stalled"**
CD and CL are inputs given to the aero-elastic modeling. How these cases are treated ? Is this a LUT of measured CL/CD or a Xfoil simulation ? Or aerodyn from fast ? Or CFD computations …?

Authors:
We have included an additional paragraph in the manuscript to clarify this point. Note that CL and CD are part of the model, implemented in FAST, and could be identified through experiments or ad-hoc calculations with Xfoil or CFD.

**Q6 p8L30: "A direct comparison between VTB and VB reveals that the latter provides … are scaled by a factor of c=0.928"**
Why the model used to compute the aerodynamic coefficients is suspected to be the source of error? Is the model used limited? Are the operating AoA in the stall region? Why not a misalignment bias of the rotor or difference of pitch angles between blades during installation?

Authors:
We have updated the manuscript to address these points. Please see also the subsequent paragraph,
which discusses the calibration with respect to the met-mast reference also shown in Fig. 4 and 5.
Note that the model is not limited to the linear regime, and can also be employed in the stall region
(see added paragraph in Section 3.5).

**Q7 p10: "possible bias in the measurement of the azimuthal position of the rotor" or "no blade dynamics included in the model"**
How the azimuth is measured, is the 11.4° in the error range of the sensor? Why you didn't include the blade dynamic model ? This would have been interesting to cross-check your hypothesis and discriminate between a sensor error or a modeling error.

Authors:
The error range of the sensors is unknown to the authors and might depend on the specific installation.
A blade dynamic model has not been used  mainly because it would require additional precise knowledge of additional model parameters, which – if not know with sufficient accuracy (as it might very often be the case) – could lead to a complex propagation of errors. As the steady blade out-of-plane bending moments of this study already do not exactly fit the model predictions without ad hoc corrections, we expect additional difficulties when relying on even more rotor parameters. Therefore, our approach was based on the principle that the formulation should not be more complicated than strictly necessary. Section 2.1 has now been expended to address and explain this choice.

**Q8 p15 figure 10:** Is it possible to have the floris pictures between instant C and instant 5:00, where there is a peak increase of velocity Vs,left? It seems to me that the rotor orientation hasn't changed much relatively to the instant C (gama is constant ~145° ) while the Vs,left peak is quite significant and the Vs,right remain constant (waked condition). This dissymmetry in the wind estimation (and therefore in load bending moments) is quite strange if the wind orientation hasn't changed. Maybe an errorbar in the measurement of the wind orientation may help?

Authors:
Please see Q10 below.

**Q9: p15 figure 10**.
Another point that is remarkable is instant ~9:00. While the wind direction is back to the level found after instant C (~149°), the deficit is not as high and the dissymmetry between Vs,left and Vs,right is again present. I suspect a too fast wind direction variation for the wake to develop. In another word, apart from errors in the method, is the wind unsteadiness can be suspected. Standard deviation of the wind direction may help to go a bit further in the analysis. I understand that without reference this is difficult to explain, however this high sensitivity to the time duration within a wind orientation is certainly to be estimated off-line with a dynamic calibration of the sensor method in future work. It should be at minimum reported or commented in the present paper.

Authors:
Please see Q10 below.

**Q10 (figure 10):** the coefficient k is interesting but not commented, why is that? The passage from a positive shear to a negative shear, the level of the shear at 5:00 compared to 9:00 etc ...

Authors:
The figure has been updated: the yaw orientation has been included, the reference turbulence intensity is indicated and the time instant D has been changed to 5:00. Note that already a few degrees in wind direction change can have a significant effect on wake position (compare the new FLORIS flow fields for C and D). The discussion of figure 10 has also been updated, addressing the questions raised by the reviewer.

**Q11 P17L5: "very few measurement points are available" induces "frequent shutdows of WT1"**
Can you be clearer ? I don't understand this logic: even if the wind turbine is stopped you should have bending moments measurement points ?

Authors:
We have not yet developed a wind estimator that would work when the machine is not in power production. In fact, we expect some difficulties at very low or null rotational speeds, when the airfoils might be operating at large angles of attack, possibly in the presence of complex dynamic separation phenomena. We have updated the paragraph to better explain this point.

**Q12 p17L10: "Fig. 11, suggests a small bias in the met-mast wind direction measurement and/ or that the wake is not developing exactly along the downstream direction."**
Also suggested by figure 10 with the dissymmetry between Vs,left, Vs,right ?

Authors:
Yes, the slightly displaced wake (caused by bias in wind direction measurement or wake displacement) in Fig. 13 (formerly 11) may be also present in Fig. 12 (formerly 10). In the latter case, this is however difficult to assess with certainty.

**Q13 P17L16: "the scatter ..."**
It can also be attributed to the level of the atmospheric turbulence in the inflow, a comparison from std from met-mast and std of Vs,right / Vs,left may help to assess this point?

Authors:
We have now included the turbulence intensity in Fig. 12 (formerly Fig. 10) and discuss there its effect on wake development.

**Q14 p18L25: "The larger fluctuations of the vertical shear compared to the horizontal one are probably caused by varying ambient inflow conditions."**
Depending on the mast instrumentation (sonic or vanes), this point can be assessed by the evaluation of the atmospheric stability and thus possible additional velocity fluctuations in the vertical direction.

Authors:
Unfortunately, we have no additional information and are unable to further comment on this aspect.

**Q15:**
**P19-20: "This indicates that some of the scatter ...proposed method"**
**P20L4: "Clearly, this is simply a feature of the flow, and not of the method tested here."**
These sentenses are very affirmative while there was no clear demonstration on that purpose. Clearly tendencies agree well with what is expected and the method gives interesting results. However, additional measurement points are needed to have an effective measure of the method accuracy in space (more points on the mast in the vertical direction, maybe a mast in the horizontal direction, and some topological analysis of the terrain …).

Authors:
We have rephrased these sentences.

**Q16 p19L4: "(…) waked by a second machine. This feature of the test site has been exploited for demonstrating the ability of the proposed local wind sensing technique to detect wake impingement."**
The measurements available on field test site is not able to perform a direct validation of the method, which would consist on a direct comparison between a full spatio-temporal description of the wind inflow (at least a 2D plan) with the estimated one. The demonstration is rather based on analysis from partially available measurements (mast, SCADA, azimuth, …) completed with wake estimation from FLORIS. More specifically, there is no way to validate the horizontal shear (wake) with inflow measurements (only one point). Tendencies are clearly coherent to what we would expect, but a precise evaluation of the method accuracy (in time and space) is not feasible. The term "demonstrating" is therefore a bit strong here, especially for the wake detection.

Authors:
We have rephrased this sentence.

Minor corrections
**C1:** In equation 1a, V should be replaced by VTB and in equation 1b, V should be replaced by Vi

Authors:
The coefficients are defined with respect to the ambient uniform wind speed (same for the tip-speed-ratio and dynamic pressure later on). VTB and Vi are estimated through Eq. (2).

**C2:** Usual conventions for wind roses representations are: North corresponds to $0°/360°$, East to $90°$, South to $180°$ and West to $270°$. In figure 2, $0°/360°$ corresponds to South.

Authors:
Indeed we follow this convention. Note that Figure 2 includes arrows, to correctly indicate wind directions.

**Referee comment 2 (RC2)**
Thank you for this paper. I apologize up front that due to school closures and work hour impacts, this will be a brief review. That said, the paper is of high quality such that I have very little in the way of criticism. The paper follows a set of earlier papers (described in the introduction) which develop the methods tested in this paper, and evaluate it in aero-elastic, LES and wind tunnel

testing. The current paper tests the estimation approaches on a full-scale test site. The results are completely convincing. The presence of the nearby met mast offers a very good validation to compare estimation of speed, shear and wake position and the analysis is clear and direct to follow, the conclusions well-justified by the presented figures. Finally, the introduction and literature are well covered, and the paper put well in the context of the broader research areas which could utilize estimation like this. I checked the equations and didn't notice any obvious errors. Recommend accepting.

Authors:
Thank you for your positive review.

**Small comments**
1) Is the cone coefficient a standard value, or an innovation of an earlier paper in this series?

Authors:
The cone coefficient was introduced in one of our previous papers. This fact has now been clarified in the updated manuscript.

2) Section 3.6: "Using again the first 7 days of measurements, the azimuth bias was identified as $\psi_{bias} = 11.4$ deg. In the remainder of this work, the sector-effective wind speeds and the two shears are computed using the corrected azimuth signal $\psi_{corr} = \psi + \psi_{bias}$."
This was interesting, as it reminds me off the offset value one might compute in the design of standard IPC controllers for 1P or 2P decoupling. Is this the same value?

Authors:
We are not familiar with the offset mentioned by the reviewer. The azimuth bias in this work is likely caused by a measurement bias and by having neglected blade dynamics (see also the discussion on this point above).

**Referee comment 3 (RC3)**
The manuscript entitled "Field testing of a local wind inflow estimator and wake detector" deals with the full-scale experimental validation of estimator concepts based on the use of the rotor as a wind sensor. The methods are based on the processing of the blade load fluctuations, and particularly the blade out-of-plane bending moments. Since 2010, the research team lead by Bottasso developed, improved and validated the concept of using the rotor as a wind sensor and the present paper is in line with this continuous process. It reaches a new step, by performing the demonstration and partial validation of the concept at full scale, on utility-scale wind turbines. The main challenges are then to obtain statistically converged, reliable and exploitable results when the boundary conditions of the experiments are non-controllable and partially known (onsite environmental and atmospheric conditions) and when the propotype is not specifically designed and equipped for R&D purpose experiments (utility-scale wind turbine). These aspects lead to the need for an extensive preparation of the database by using massive data pre-processing (ad-hoc calibrations and corrections, sample rejections, filtering, classification, etc.). In the present paper, these unavoidable pre-processing steps, as well as the actual data processing steps, are well argued,

described and illustrated. The obtained results are on general, well explained and prove the feasibility of the "WT as a wind sensor" concept. On the other hand, a lack of information on the experimental set-up and on the site description affect sometimes the reliability of the results interpretation, leading the authors to use too frequently "likely", would", "seems to", could be due to". Mainly, a better knowledge of the site properties (terrain and micrometeorology) can help in some interpretations. This can be provided a posteriori using geographical and meteorological databases and it is essential to add them to the manuscript.

Authors:
Thank you for your positive review and helpful comments. We have expanded the manuscript by adding more information on the experimental set-up and by improving the site description.

Major comments:
-   A thorough description of the experimental set-up must be added: measurement device (anemometers, strain gages, etc.) descriptions (type, brand, accuracy, cut-off frequency, etc.)

Authors:
Please see the updated manuscript and our reply to the general comments of RC1.

-   A thorough description of the site properties must be added: type of terrain surrounding the site (type of vegetation, associated roughness length), atmospheric boundary layer properties (wind rose, averaged power law and turbulence intensity at hub height for the studied wind directions, thermal stability encountered during the selected periods, etc.). If not findable by the measurement campaign itself, meteorological information can be extracted from global meteorology reanalysis database as MERRA2 or ERA5.

Authors:
We have included additional information in the manuscript and refer to Bromm 2017, which describes the same site with reference to another research study. We believe that the new description is sufficient for the scope of this work.

-   §3.3 Reference inflow & Figure 3 : it is written that the wind speed is measured at three different heights on the met-mast but two of them are located at 2m of each other. Therefore, one cannot consider that one has three distinct values to assess the power law exponent, but only two. What is the consequence on the accuracy of the obtained power law exponent?

Authors:
The power law exponent can be estimated using only 2 measurement heights, as it depends on only two free parameters. The two measurements located at 2m of each other could be combined into one mean measurement without a significant change in the power law exponent estimation. This point has been added to the manuscript.

-   Figures 7 and 8: The obtained values for the power law exponent (mainly between 0.2 and 0.4) are particularly high for such an open-field terrain, as it seems to be on the satellite picture. These values are usually encountered on rough to very rough terrains (forest or

city). Again, a better description of the terrain fetch and of the local atmospheric boundary layer properties would help to justify the results reliability.

Authors:
The site is not completely flat and open (see updated description). We also added a summary of the atmospheric conditions that were reported in Bromm 2017.

- Page 12, lines 9-10: "This difference could possibly be caused by a non-ideal power law inflow profile, leading to a biased met-mast reference shear, although a definitive explanation of this mismatch could not be reached with the present data set.". I would recommend to make a sensitivity analysis on the power law exponent to the number and position of the used anemometers

Authors:
Please see the comment before the previous one. Also note that we include a reference to Møller 2020, which nicely shows such non-ideal power law profiles.

- Pages 12-13: "Considering that all wind directions are for un-waked met-mast and turbine, these results suggest the presence of a spatial shear variation, probably caused by the local vegetation." Again, a better description of the terrain fetch and of the local atmospheric boundary layer properties would help to justify this assumption.

Authors:
The description of the site has been expanded, and this sentence has been updated.

- It is written on page 8, lines 4-5, "Measurements taken during yawing manoeuvres were also discarded, as additional induced loads can pollute the estimates". On the other hand, on Figures 6 and 10, the wind direction progressively changes from 240_ to 200_during 6 hours, and from 100_ to 175] in 12 hours, respectively. Yaw manoeuvres should appear during these periods. It sounds in opposition of the first statement. Could you please add the wind turbine orientation time series to these plots and explain how you did the data analysis during these periods?

Authors:
We updated the description on how the consecutive 10-min averages have been computed, and provided a clearer explanation of the effect of yaw maneuvers.

- Figure 11 : would it be possible to classify the results considering the incoming wind speed category (and so the wind turbine operating point). One could expect that the wake is more or less intense, depending on the wind turbine operating point and that the wake detector is more or less efficient.

Authors:
Below you can find a classification of the results in terms of turbulence intensity (higher turbulence should enhance wake recovery and therefore lead to less intense but wider wakes) and in terms of wind speed (note that there are almost no data points above rated wind speed for these wind directions). However, conclusions are difficult to obtain, perhaps due to the very limited amount of data.

As a side-note, please notice that the sign of the horizontal shear is now the opposite of the one of the original manuscript, for consistency with other publications.

[Figure]

- Page 17, lines 10-12: "the wake is not developing exactly along the downstream direction. Indeed, the latter is a well-known phenomenon observed in vertically sheared flow (Vollmer et al., 2016)." Yes, it is true for yawed wind turbines, or for un-yawed ones in very stable atmospheric conditions but cannot be considered as a universal explanation for the bias in the present results.

Authors:
We updated the text.

- Page 18, lines 24-25: "The larger fluctuations of the vertical shear compared to the horizontal one are probably caused by varying ambient inflow conditions." It is not clear what this sentence means exactly. Could you elaborate more on these "varying ambient inflow conditions"? Again, a better knowledge of the local atmospheric boundary layer properties can help to justify some results.

Authors:
We rephrased this sentence.

- Conclusions: some conclusions are not new (i.e. "rotor-effective wind speed can be estimated from blade out-of-plane bending moments, with a quality that is nearly indistinguishable from the well-known torque-balance method"), since already drawn by previous papers from the same research team. What is new is to make the full-scale demonstration/validation of these different concepts.

Authors:
We have reformulated the conclusions.

**Minor comments**
- Page 3, line 17: remove A in the q formula

Authors:
Thank you.

- Page 4, line 3-4: "A rotor-effective wind speed can also be obtained from the blade-effective ones by simple averaging over all (three) blades". One expects that the dynamics of the rotoreffective speed is quite low (cut-off frequency linked to the rotor diameter, whereas the dynamics of the blade-effective ones must be higher. Do you get the right rotoreffective speed dynamics by averaging the three blade-effective wind speeds?

Authors:
The polar inertia of the rotor is certainly much higher than the flap inertia of each blade, so it is indeed possible that the two estimators have different cutoff frequencies. However, we have not investigated this point in detail, nor this comparison between the two methods is very relevant for the scope of the present paper. Also note the new section "3.3 Estimator update frequency", which shows that the wind estimation based on flapwise loads is capable of following relatively fast fluctuations of the inflow.

- Page 4, line 17-18 "he smaller inertia and high damping of this degree of freedom makes this more sophisticated approach superfluous": Please add a reference to prove this statement.

Authors:
We have updated this paragraph.

- Page 5, figure 1: the reference framework (x,y,z) is not direct. Considering the naming convention in the downstream viewing direction, one assumes that x is in the downstream direction too. Then y, should be oriented on the left

Authors:
We have updated the reference frame. Note that we also changed the horizontal shear definition for consistency with other publications.

- Page 9, lines 7-8: add this information into the experimental set-up description

Authors:
We have updated the experimental set-up description.

- Figures 4& 5: should be written in the captions that it is after correction

Authors:
We have updated the caption of Fig. 6 (formerly 5) only as the V_TB in Fig. 5 (formerly 4) is not affected by corrections.

Revised version of the manuscript with highlighted changes

[revised manuscript text omitted]